# Decreased recent adaptation at human mendelian disease genes as a possible consequence of interference between advantageous and deleterious variants

**Chenlu Di[1], Jesus Murga Moreno[2], Diego F Salazar-Tortosa[1], M Elise Lauterbur[1], David Enard[1]***

[1]University of Arizona Department of Ecology and Evolutionary Biology, Tucson, United States; [2]Institut de Biotecnologia i de Biomedicina and Departament de Genètica i de Microbiologia, Universitat Autònoma de Barcelona, Barcelona, Spain

**Abstract** Advances in genome sequencing have improved our understanding of the genetic basis of human diseases, and thousands of human genes have been associated with different diseases. Recent genomic adaptation at disease genes has not been well characterized. Here, we compare the rate of strong recent adaptation in the form of selective sweeps between mendelian, non-infectious disease genes and non-disease genes across distinct human populations from the 1000 Genomes Project. We find that mendelian disease genes have experienced far less selective sweeps compared to non-disease genes especially in Africa. Investigating further the possible causes of the sweep deficit at disease genes, we find that this deficit is very strong at disease genes with both low recombination rates and with high numbers of associated disease variants, but is almost non-existent at disease genes with higher recombination rates or lower numbers of associated disease variants. Because segregating recessive deleterious variants have the ability to interfere with adaptive ones, these observations strongly suggest that adaptation has been slowed down by the presence of interfering recessive deleterious variants at disease genes. These results suggest that disease genes suffer from a transient inability to adapt as fast as the rest of the genome.

**\*For correspondence:**
denard@email.arizona.edu

**Competing interests:** The authors declare that no competing interests exist.

## Introduction

Advances in genome sequencing have dramatically improved our understanding of the genetic basis of human diseases, and thousands of human genes have been associated with different diseases (*Amberger et al., 2019*; *Piñero et al., 2020*). Despite our expanding knowledge of mendelian disease gene associations, and despite the fact that multiple evolutionary processes might connect disease and genomic adaptation at the gene level, these connections are yet to be studied more thoroughly, especially in the case of recent genomic adaptation. Different evolutionary processes have the potential to make the occurrence of mendelian disease genes and adaptation not independent from each other in the human genome. For instance, hitchhiking of deleterious mutations linked to advantageous mutations might increase the risk of mendelian disease-causing variants at genes subjected to past directional adaptation. Disease genes might then appear to have experienced more adaptation than non-disease genes if this specific process was sufficiently widespread. Conversely, higher evolutionary constraint, and higher pleiotropy might reduce adaptation at disease genes compared to genes not involved in diseases (*Otto, 2004*). There is currently considerable uncertainty about how any of these non-exclusive evolutionary processes, or other processes, might have influenced adaptation at disease genes. It is even not well-known whether human non-

infectious disease genes have similar, higher or lower levels of adaptation in human populations compared to genes not involved in diseases. Comparing levels of adaptation between disease genes and non-disease genes is a first important step toward better understanding the evolutionary relationship between non-infectious diseases and genomic adaptation.

Multiple recent studies comparing evolutionary patterns between human mendelian disease and non-disease genes have found that mendelian disease genes are more constrained and evolve more slowly (*Blekhman et al., 2008*; *Quintana-Murci, 2016*; *Spataro et al., 2017*; *Torgerson et al., 2009*). An older comparison by *Smith and Eyre-Walker, 2003* found that disease genes evolve faster than non-disease genes, but we note that the sample of disease genes used at the time was very limited.

The significant increase of the number of known disease genes since these studies were completed makes it important to update the comparison of evolutionary patterns at disease and non-disease genes. More critically however, past studies all have in common an important limitation that justifies comparing disease genes and non-disease genes again. Disease and non-disease genes may differ by more than just the fact that they have been associated with disease or not. Disease and non-disease genes may also differ in many other factors other than their disease status. Such factors can be a problem when comparing adaptation in disease genes and non-disease genes, because they, instead of the disease status itself, could explain differences in adaptation. For example,

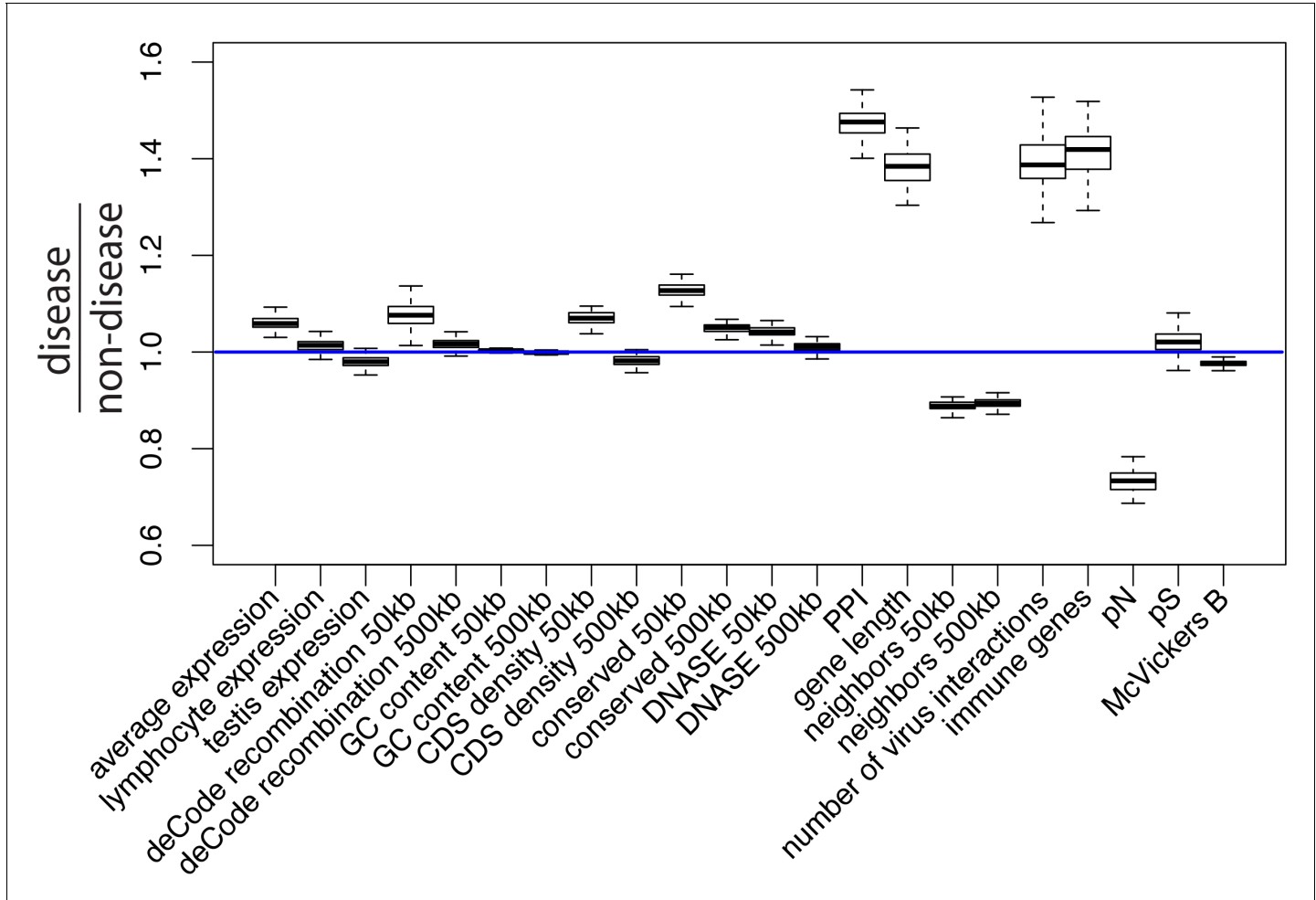

**Figure 1.** Potential confounding factors in disease versus non-disease genes. Each potential confounding factor is detailed in the Materials and methods. For each confounding factor, the boxplot shows on the y-axis the ratio of the average factor value for disease genes, divided by the average factor value for non-disease genes. The boxplot error bars are obtained by calculating the ratio 1000 times, each time by randomly sampling as many non-disease genes as there are disease genes.

disease genes tend to be more highly expressed than non-disease genes (*Spataro et al., 2017*; *Figure 1*). If higher expression happens to be associated with more adaptation in general, one might detect more adaptation in disease genes in a way that has nothing to do with disease, and just reflects their higher levels of expression. Many other factors may also be important. For example, immune genes, which often adapt in response to infectious pathogens, may further complicate comparisons if they are represented in unequal proportions between non-infectious disease and non-disease genes. Comparing genomic adaptation in disease and non-disease genes thus requires careful consideration of confounding factors.

Among possible confounding factors, it is particularly important to take into account evolutionary constraint, that is the level of purifying selection experienced by different genes. A common intuition is that mendelian disease genes may exhibit less adaptation because they are more constrained (*Blekhman et al., 2008*; *Spataro et al., 2017*; *Torgerson et al., 2009*), leaving less mutational space for adaptation to happen in the first place. Less adaptation at mendelian disease genes might thus represent a trivial consequence of varying constraint between genes (*Kim et al., 2007*), which says little about a specific connection between disease and adaptation. In the same vein, one might expect disease genes to be associated with higher mutation rates, and more frequent adaptation to follow as a trivial consequence of elevated mutation rates. Whether disease genes experience higher mutation rates is however still an open question (*Eyre-Walker and Eyre-Walker, 2014*; *Osada et al., 2009*). In any case, focusing specifically on disease and adaptation requires controlling for confounders such as constraint/purifying selection and mutation rate (see Materials and methods, Results and *Figure 1* for a complete list of confounders accounted for in this analysis).

A specific evolutionary relationship may exist between adaptation and disease beyond the simple effect of constraint, mutation rate or other confounders. In an evolutionary context, once constraint and other confounding factors have been accounted for, we can imagine three potential scenarios for the comparison of adaptation between disease and non-disease genes. Under scenario 1, any potential difference in adaptation between disease and non-disease genes is entirely due to differences in constraint and other confounding factors. Under this scenario, there is no further evolutionary process linking disease and adaptation together. Therefore, there is no difference in adaptation between disease and non-disease genes once confounding factors have been accounted for.

Under scenario 2, always once selective constraint and other confounding factors have been accounted for, disease genes have more adaptation than non-disease genes. For example, as already mentioned above, deleterious mutations can hitchhike together with adaptive mutations to high frequencies in human populations (*Barreiro and Quintana-Murci, 2010*; *Birky and Walsh, 1988*; *Chun and Fay, 2011*; *Quintana-Murci and Barreiro, 2010*). Other, less well established, cases can be imagined where past adaptation decreased the robustness of a specific gene, and subsequent mutations become more likely to be associated with diseases (*Xu and Zhang, 2014*). Scenario 2 thus favors a relationship between adaptation and disease, where past adaptation precedes and influences the likelihood of a gene being associated with disease.

Under scenario 3, disease genes have less adaptation than non-disease genes even after accounting for confounding factors such as evolutionary constraint. Such a scenario might occur for example if disease genes happen to be genes that can be sensitive to changes in the environment, with a fitness optimum that can change over time, but where adaptation has not occurred yet to catch up with the new optimum. Such an adaptation lag (or lag load, to reuse the terminology introduced by *Maynard Smith, 1976*) may occur for example if higher pleiotropy at disease genes (*Ittisoponpisan et al., 2017*) makes it less likely for new mutations to be advantageous (*Otto, 2004*) (in addition to increasing the level of constraint already accounted for as a confounding factor). An adaptation lag may also occur if deleterious mutations interfere with and slow down adaptation at disease genes more than at non-disease genes (*Assaf et al., 2015*; *Hill and Robertson, 1966*). Alternatively, disease genes could have constitutively less adaption, because of pleiotropy or because new mutations tend to be large effect mutations that often overshoot the fitness optimum, which would prevent them from being advantageous. In the latter scenario of a constitutive deficit of adaptation, disease genes should have not only a deficit of recent adaptation, but also a deficit of older, long-term adaptation, that can be estimated with approaches such as the McDonald-Kreitman test (*McDonald and Kreitman, 1991*).

Even though uncovering the underlying evolutionary processes that govern the relationship between disease and adaptation will take a lot more work, it is important to find first which scenario

is the most likely to be true, that is whether disease genes have as much, more, or less adaptation than non-disease genes. Finding out which out of the three possible scenarios is true may give a preliminary basis to further hypothesize which evolutionary processes are more likely to dominate the relationship between disease and adaptation genome-wide.

Here, we compare recent adaptation in mendelian disease and non-disease genes in order to disentangle the connections between adaptation and disease. We specifically compare the abundance of recent selective sweeps signals, where hitchhiking has raised haplotypes that carry an advantageous variant to higher frequencies (*Smith and Haigh, 1974*). Note that this means that we can only compare adaptation at specific loci between disease and non-disease genes that was strong enough to induce hitchhiking, hence we do not take into account polygenic adaptation distributed across a large number of loci that did not leave any hitchhiking signals (see Discussion). As mentioned above, confounding factors may affect the comparison between disease and non-disease genes. In contrast with previous studies, we systematically control for a large number of confounding factors when comparing recent adaptation in human mendelian disease and non-disease genes, including evolutionary constraint, mutation rate, recombination rate, the proportion of immune or virus-interacting genes, etc. (please refer to Materials and methods for a full list of the confounding factors included). In addition to controlling for a large number of confounding factors, we estimate false positive risks (FPR) (*Colquhoun, 2019*) for our comparison pipeline that fully take into account the implications of controlling for many confounding factors (see Materials and methods and Results).

As a list of mendelian disease genes to test, we curate human mendelian non-infectious disease genes based on annotations in the DisgeNet (*Piñero et al., 2020*) and OMIM (*Amberger et al., 2019*) databases (Materials and methods). We focus on non-infectious mendelian disease genes rather than all disease genes including complex disease associations, because different evolutionary patterns can be expected between mendelian and complex disease genes based on previous studies (*Blekhman et al., 2008*; *Quintana-Murci, 2016*; *Spataro et al., 2017*; *Torgerson et al., 2009*). In total, we compare 4215 mendelian disease genes with non-disease genes in the human genome. In agreement with scenario 3, we find a strong deficit of selective sweeps at disease genes compared to non-disease genes in Africa, and weaker deficits in East Asia and Europe. We further test multiple potential explanations for this deficit pattern across human populations, and find that it strongly depends on recombination and the number of known disease variants at given mendelian disease genes. This suggests that segregating deleterious mutations at disease genes might interfere with, and slow down genetically linked adaptive variants enough to produce the observed lack of sweeps at disease genes. We further support this possible explanation with forward population simulations (*Haller and Messer, 2019*) that show that stronger interference in Africa compared to East Asia or Europe is expected. We also show that alternative explanations implying an evolutionarily stable, constitutive deficit of advantageous mutations, rather than transient interference, are made unlikely by the fact that although disease genes have experienced less recent adaptation in the form of sweeps, they have not experienced less long-term adaptation during the millions of years of human evolution.

## Results

### Controlling for confounding factors with a bootstrap test

To compare mendelian disease and non-disease genes, we first ask which potential confounding factors differ between the two groups of genes. As expected, multiple measures of selective constraint are significantly higher in mendelian disease compared to non-disease genes. As a measure of long-term constraint, the density of conserved elements across mammals is slightly higher at disease genes compared to non-disease genes (*Figure 1*: conserved 50 kb, conserved 500 kb; Materials and methods).

As a measure of more recent constraint, we contrast pS, the average proportion of variable synonymous sites, with pN, the average proportion of variable nonsynonymous sites (*Figure 1*; Materials and methods). If the coding sequences of disease genes are more constrained, we expect a drop of pN at disease genes, but no such drop of pS at neutral synonymous sites. Accordingly, pN is lower at disease compared to non-disease genes, while pS is very similar between the two

categories of genes (*Figure 1*). Therefore, selective constraint was stronger in the coding sequences of disease genes during recent human evolution.

As another measure of recent constraint, we also use McVicker's B estimator of background selection (*McVicker et al., 2009*). The amount of background selection at a locus can be used as a proxy for recent constraint, since it depends on the number of deleterious mutations that were recently removed at this locus. The lower B, the more background selection there is at a specific locus. In line with higher recent constraint at disease genes, B is slightly, but significantly lower at disease genes (*Figure 1*; Materials and methods). Overall, we find evidence of higher constraint at disease genes.

These differences between disease and non-disease genes highlight the need to compare disease genes with control non-disease genes with similar levels of selective constraint. To do this and compare sweeps in mendelian disease genes and non-disease genes that are similar in ways other than being associated with mendelian disease (as described in the Results below, Less sweeps at mendelian disease genes in Africa), we use sets of control non-disease genes that are built by a bootstrap test to match the disease genes in terms of confounding factors (Materials and methods), including the confounding factors that represent measures of selective constraint/purifying selection (density of conserved elements, pN and pS, and McVicker's B; see Materials and methods). To verify that the measures of selective constraint included indeed control for purifying selection when comparing disease and matched control non-disease genes, we ran a maximum likelihood version of the McDonald-Kreitman test called GRAPES (*Galtier, 2016*), to compare the average selection coefficient of deleterious mutations at disease gene coding sequences, compared to the control non-disease gene coding sequences (Materials and methods). We find that disease genes and their non-disease control genes have undistinguishable average strengths of deleterious variants, suggesting that our controls for selective constraint are sufficient, at least to account for constraint at the coding sequence level (*Figure 2*; comparison test p=0.37). Further down in the Results (Verification of purifying selection controls), we also show that we properly control for purifying selection not just in coding sequences but in the whole genomic regions that we analyze by using GERP (*Davydov et al., 2010*).

In addition to constraint, mutation rate could represent an important confounder. The proportion of variable neutral synonymous sites pS can be used to compare mutation rates, since the number of variable synonymous sites is proportional to the mutation rate under neutrality. As mentioned

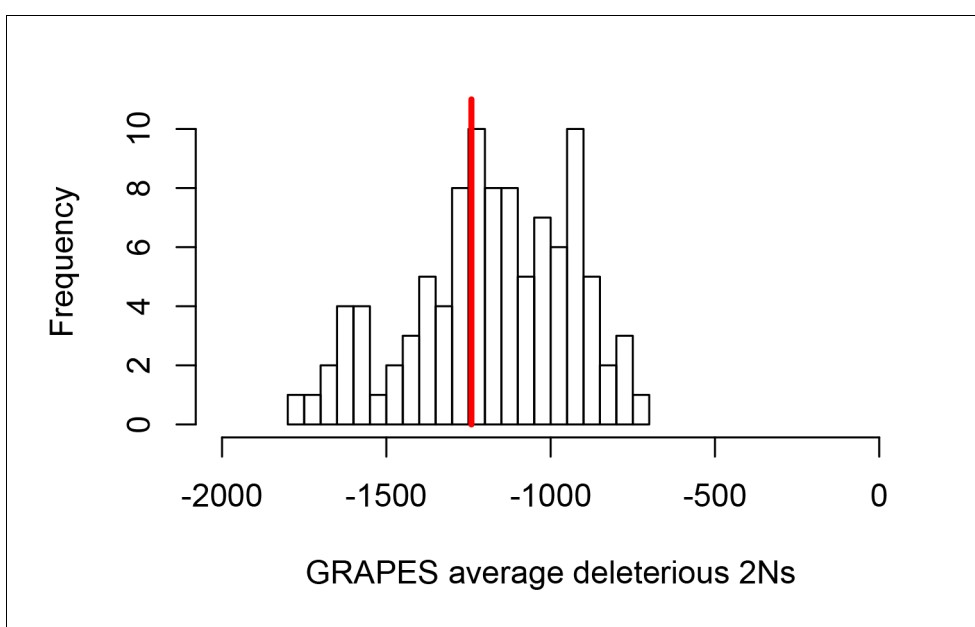

**Figure 2.** Average strength of deleterious nonsynonymous variants in disease vs control genes. The average strength of deleterious nonsynonymous variants was measured using GRAPES with the DisplGamma distribution of fitness effects, which gave the best fit to disease and control sets. The histogram represents 100 control sets. The red line represents the average strength of deleterious nonsynonymous variants in mendelian disease genes (2Ns=-1241).

already, pS is very similar at disease and non-disease genes (*Figure 1*), suggesting that mutation rates are similar at disease and non-disease genes. This is further supported by the fact that multiple factors that could affect the mutation rate such as GC content or recombination are also similar or very slightly different at disease and non-disease genes (*Figure 1*; Materials and methods). Aside from mutation rate and constraint, multiple other factors that could affect adaptation differ between disease and non-disease genes, notably including the proportion of genes that interact with viruses, the proportion of immune genes, or the number of protein-protein interactions (PPIs) in the human PPIs network. All these factors have been shown to, or could in principle affect adaptation (Materials and methods), further showing the necessity to control for confounding factors when comparing adaptation at disease and non-disease genes. The fact that previous studies comparing adaptation at disease versus non-disease genes did not control for confounding factors, makes it unclear if their conclusions reflect properties tied to genes being associated with disease or not, or tied to other confounding factors not accounted for.

## Less sweeps at mendelian disease genes in Africa

For our comparison of disease and non-disease genes, we measure recent adaptation around human protein coding genes (Materials and methods) using the integrated Haplotype Score iHS (*Voight et al., 2006*) and the number of Segregating sites by Length nSL (*Ferrer-Admetlla et al., 2014*) in 26 populations rom the 1,000 Genomes Project (*Auton et al., 2015*) (Materials and methods). The iHS and $nS_L$ statistics are both sensitive to recent incomplete sweeps, and have the advantage over other sweep statistics of being insensitive to the confounding effect of background selection (*Enard et al., 2014*; *Schrider, 2020*). To evaluate the prevalence of sweeps at disease genes relative to non-disease genes, we do not use the classic outlier approach, and instead use a previously described, more versatile approach based on block-randomized genomes to estimate

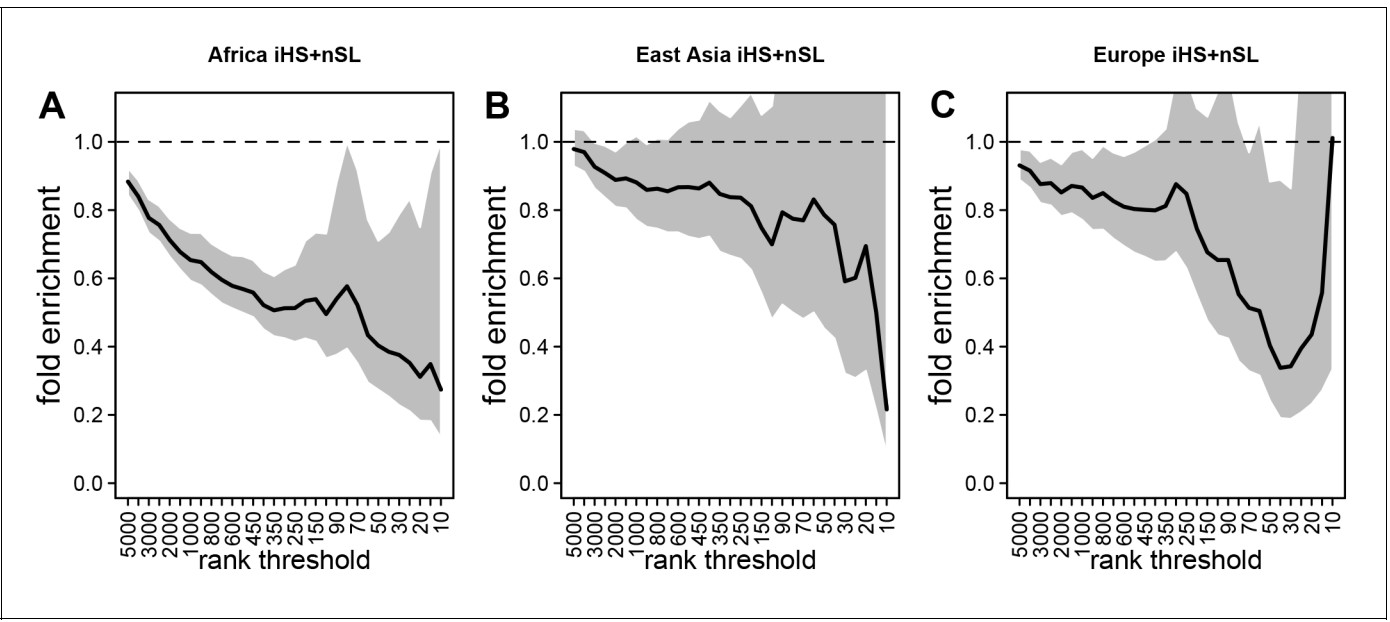

**Figure 3.** Deficit of iHS and $nS_L$ sweep signals at mendelian disease genes. The figure shows the averaged whole enrichment curves and their averaged confidence intervals from the bootstrap test, averaged over both iHS and $nS_L$ sweep ranks, and over all the populations from each continent (Materials and methods). The y-axis represents the relative sweep enrichment at disease genes, calculated as the number of disease genes in putative sweeps, divided by the number of control non-disease genes in putative sweeps. The gray areas are the 95% confidence interval for this ratio. The number of genes in putative sweeps is measured for varying sweep rank thresholds. For example, at the top 100 rank threshold, the relative enrichment is the number of disease genes within the top 100 genes with the strongest sweep signals (either according to iHS or $nS_L$), divided by the number of control non-disease genes within the top 100 genes with the strongest sweep signals. We use genes ranked by iHS or $nS_L$ using 200 kb windows, since 200 kb is the intermediate size of all the window sizes we use (50 kb, for the smallest, 1000 kb for the largest; see Materials and methods). (**A**) Africa, average over the ESN, GWD, LWK, MSL, and YRI populations from the 1000 Genomes Project. (**B**) East Asia, average over the CDX, CHB, CHS, JPT, and KHV populations. (**C**) Europe, average over the CEU, FIN, GBR, IBS, and TSI populations.

unbiased false positive risks (FPR) for whole enrichment curves (*Figure 3*; *Enard and Petrov, 2020*). We first rank genes based on the average iHS or $nS_L$ in genomic windows centered on genes (Materials and methods), from the top-ranking genes with the strongest sweep signals to the genes with the weakest signals. We then slide a rank threshold from a high rank value to a low rank value (from top 5000 to top 10, x-axis on *Figure 3*). For each rank threshold, we estimate the sweep enrichment (or deficit) at disease relative to non-disease genes (*Figure 3*, y-axis). For example, for rank threshold 200, the relative enrichment (or deficit) is the number of disease genes in the top 200 ranking genes, divided by the number of control non-disease genes in the top 200. By sliding the rank threshold, we estimate a whole enrichment curve that is not only sensitive to the strongest sweeps but also to weaker sweeps signals (for example using the top 5000 threshold; *Figure 3*). Using block-randomized genomes (Materials and methods), we can then estimate an unbiased false positive risk (FPR) for the whole enrichment curve. In brief, we estimate FPRs by re-running the entire enrichment analysis pipeline many times, but each time on a block-randomized genome instead of the real genome. Block randomized genomes are genomes where gene coordinates have been randomly shuffled in a way that preserves the original genes/sweeps clustering structure. The FPR can then be calculated by comparing the whole sweep deficit/enrichment curve in the real genome with the many observed whole sweep deficit/enrichment curves observed with the block-randomized genomes (see Materials and methods for more details). This strategy makes less assumptions on the expected strength of selective sweeps. The approach also makes it possible to estimate a single false positive risk based on the cumulated enrichment (or deficit) over multiple whole enrichment curves (Materials and methods). Here, we estimate a single false positive risk for both iHS and $nS_L$ curves considered together, and also for multiple window sizes to measure average iHS and $nS_L$ (from 50kb to 1Mb, Materials and methods).

To control for confounding factors (*Figure 1*), we compare sweep signals at disease genes with control sets of non-disease genes that were chosen by a bootstrap test (*Enard and Petrov, 2020*) because they match disease genes in terms of confounding factor values (Materials and methods). Furthermore, control non-disease genes are chosen far from disease genes (>300 kb; Materials and methods). We do this to avoid choosing as controls non-disease genes that are too close to disease genes and thus likely to have the same sweep profile (especially in the case of large sweeps potentially overlapping both neighboring disease and non-disease genes). This, together with the large number of confounding factors that we match, tends to limit the pool of possible control genes (Materials and methods). The statistical impact of a limited control pool is however fully taken into account by the estimation of a FPR with block-randomized genomes (Materials and methods).

Because they have experienced different demographic histories, we test different human populations from distinct continents separately. Specifically, we test African populations, East Asian populations and European populations from the 1000 Genomes Project phase 3 (*Auton et al., 2015*). At this stage, we must consider the fact that most gene-disease associations in our dataset were likely discovered in European cohorts. Because disease genes in Europe may not always be disease genes in other populations, we cannot exclude the possibility that a sweep enrichment or a sweep deficit might be more pronounced in Europe, unless the evolutionary processes that make a gene more likely to be a disease gene predated the split of different human populations.

Using both iHS and $nS_L$ sweep signals, we find a strong depletion in sweep signals at disease genes, especially in Africa (*Figure 3A*) compared to East Asia or Europe (*Figure 3B, C*, respectively). *Figure 3A, B, C* show the sweep deficit curves at disease genes compared to control non-disease genes in Africa, East Asia, and Europe, respectively. The corresponding false positive risks that quantify how unexpected the downward or upward skew of these curves are (Materials and methods), show that the sweep deficit is strongly significant in Africa, marginally so in Europe, and not significant at all in East Asia (FPR=$3.10^{-4}$ in Africa vs. 0.18 in East Asia and 0.05 in Europe, *Figure 4A, B C* respectively; Materials and methods). Note that this FPR takes the clustering of multiple genes in the same sweeps into account (*Enard and Petrov, 2020*). A stronger depletion in Africa suggests that the evolutionary processes linking disease and adaptation at the gene level predate the split of African and European populations, given that most gene-disease associations studies involved European cohorts. As we show below, the stronger sweep depletion in Africa can be explained in the evolutionary context of genetic interference between advantageous and deleterious variants at mendelian disease genes.

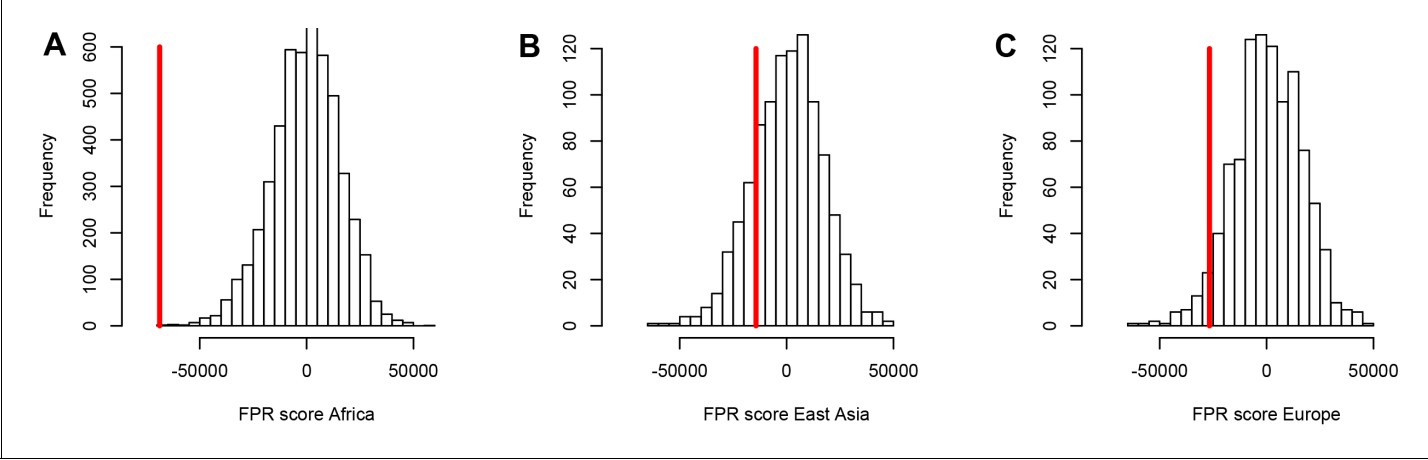

**Figure 4.** A stronger sweep deficit at disease genes in Africa than in East Asia and Europe. The figure shows the observed sweep enrichment/deficit score used to measure the false positive risk (FPR) in the real genome (red line), compared to the expected null distribution of the score estimated with block-randomized genomes (5000 block-randomized genomes in Africa, 1000 in East Asia and Europe; Materials and methods). The FPR score is based on summing the difference between the number of genes in sweeps at disease genes and the number of genes in sweeps in control genes, over both iHS and $nS_L$, and different window sizes (Materials and methods). (A) FPR score in Africa, estimated summing over the ESN, GWD, LWK, MSL, and YRI populations from the 1000 Genomes Project. (B) FPR score in East Asia, estimated summing over the CDX, CHB, CHS, JPT, and KHV populations. (C) FPR score in Europe, summing over the CEU, FIN, GBR, IBS, and TSI populations.

The online version of this article includes the following figure supplement(s) for figure 4:

**Figure supplement 1.** FPR with or without controlling for GERP.

Notably, the stronger depletion observed in Africa likely excludes the possibility that it could be mostly due to a technical artifact, where sweeps themselves might make it harder to identify disease genes in the first place. Sweeps increase linkage disequilibrium (LD) in a way that could make it more difficult to assign a disease to a single gene in regions of the genome with high LD and multiple genes genetically linked to a disease variant. This could result in a depletion of sweeps at monogenic disease genes, simply because disease genes are less well annotated in regions of high LD. However, if this was the case, because most disease gene were identified in Europe, we would expect such an artifact to deplete sweeps at disease genes primarily in Europe, not in Africa. This artifact is also very unlikely due to the fact that recombination rates are only very slightly different between disease and non-disease genes (*Figure 3*). Overall, these results support the third scenario where evolutionary processes decrease recent adaptation at mendelian disease genes. That said, it is important to note that we only detect a deficit of recent adaptation strong enough to leave hitch-hiking signals. Our results do not imply that the same is true for adaptation that is too polygenic to leave signals detectable with iHS or $nS_L$. Note that the sweep deficit at disease genes in Africa is robust to differences in gene functions between disease and non-disease genes according to a Gene Ontology analysis (Materials and methods) (*Gene Ontology Consortium and Gene Ontology, 2021*).

## Verification of purifying selection controls

To further verify that constraint/purifying selection is properly controlled for when comparing mendelian disease and control non-disease genes, we also add the GERP score, as well as the density of both coding and non-coding conserved elements identified by GERP (*Davydov et al., 2010*) to the list of matched confounding factors (Materials and methods). The average GERP score in a genomic window estimates the amount of substitutions that never happened during long-term evolution because the said mutations were removed by purifying selection (both in coding and non-coding sequences). The sweep deficit in Africa at disease genes compared to controls is completely unchanged when using GERP or not (*Figure 4—figure supplement 1*). This shows that the measures of selective constraint already included (Materials and methods) are sufficient to control for selective constraint/purifying selection. For this reason, we do not use GERP further (as explained in the

Materials and methods, the larger the number of confounding factors that we match, the lower the power of our approach to detect a sweep enrichment or deficit).

## Disease genes do not experience constitutively less long-term adaptive mutations

A deficit of strong recent adaptation (strong enough to affect iHS or $nS_L$) raises the question of what creates the sweep deficit at disease genes. As already discussed, purifying selection and other confounding factors are matched between disease genes and their controls, which excludes that these factors alone could possibly explain the sweep deficit. Purifying selection alone in particular cannot explain this result, since we find evidence that it is well matched between disease and control genes (*Figure 4* and *Figure 4—figure supplement 1*). Furthermore, we find that the 1000 genes in the genome with the highest density of conserved elements do not exhibit any sweep deficit (bootstrap test + block-randomized genomes FPR=0.18; Materials and methods). Association with mendelian diseases, rather than a generally elevated level of selective constraint, is therefore what matters to observe a sweep deficit. What then might explain the sweep deficit at disease genes?

As mentioned in the introduction, it could be that mendelian disease genes experience constitutively less adaptive mutations. This could be the case for example because mendelian disease genes tend to be more pleiotropic (*Otto, 2004*), and/or because new mutations in mendelian are large effect mutations (*Quintana-Murci, 2016*) that tend to often overshoot the fitness optimum, and cannot be positively selected as a result. Regardless of the underlying processes, a constitutive tendency to experience less adaptive mutations predicts not only a deficit of recent adaptation, but also a deficit of more long-term adaptation during evolution. The iHS and nSL signals of recent adaptation we use to detect sweeps correspond to a time window of at most 50,000 years, since these statistics have very little statistical power to detect older adaptation (*Sabeti et al., 2006*). In contrast, approaches such as the McDonald-Kreitman test (MK test) (*McDonald and Kreitman, 1991*) capture the cumulative signals of adaptative events since humans and chimpanzee had a common ancestor, likely more than 6 million years ago.

To test whether mendelian disease genes have also experienced less long-term adaptation, in addition to less recent adaptation, we use the MK tests ABC-MK (*Uricchio et al., 2019*) and GRAPES (*Galtier, 2016*) to compare the rate of protein adaptation (advantageous amino acid changes) in mendelian disease gene coding sequences, compared to confounding factors-matched non-disease controls (Materials and methods). We find that overall, disease and control non-disease genes have experienced similar rates of protein adaptation during millions of years of human evolution, as shown by very similar estimated proportions of amino acid changes that were adaptive (*Figure 5A, B,C,D,E*). This result suggests that disease genes do not have constitutively less adaptive mutations. This implies that processes that are stable over evolutionary time such as pleiotropy, or a tendency to overshoot the fitness optimum, are unlikely to explain the sweep deficit at disease genes. If disease genes have not experienced less adaptive mutations during long-term evolution then the process at work during more recent human evolution has to be transient, and has to has to have limited only recent adaptation. It is also noteworthy that both disease genes and their controls have experienced more coding adaptation than genes in the human genome overall (*Figure 5A*), especially more strong adaptation according to ABC-MK (*Figure 5C*). The fact that the baseline long-term coding adaptation is lower genome-wide, but similarly higher in disease and their control genes, also shows that the matched controls do play their intended role of accounting for confounding factors likely to affect adaptation. The fact that long-term protein adaptation is not lower at disease genes also excludes that purifying selection alone can explain the sweep deficit at disease genes, because purifying selection would then also have decreased long-term adaptation. A more transient evolutionary process is thus more likely to explain our results.

## A possible role of interference of deleterious mutations

The underlying evolutionary process at mendelian disease genes must explain the sweep deficit, while simultaneously not implying a long-term deficit of adaptation. A possible explanation is that adaptation may be limited at disease genes due to currently segregating deleterious mutations interfering with, and slowing down advantageous variants. This process may in principle satisfy the condition of decreasing recent adaptation without decreasing long-term adaptation, since the

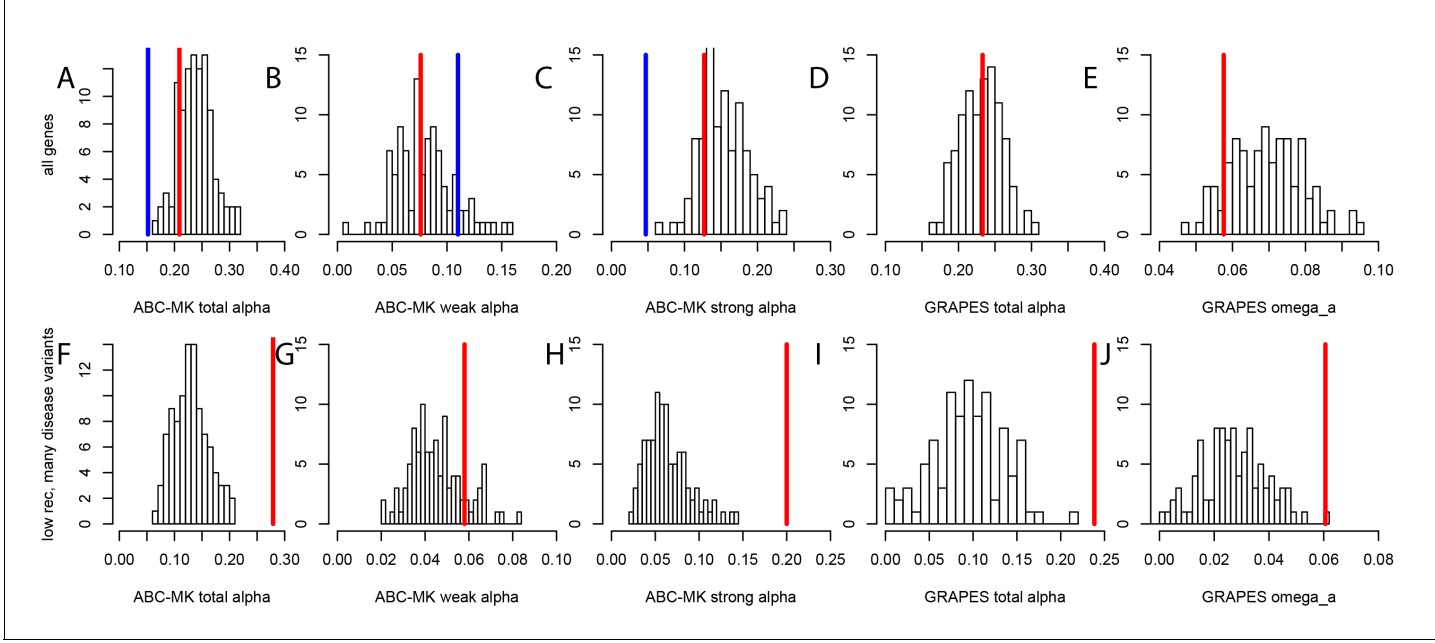

**Figure 5.** Nonsynonymous coding adaptation in disease vs. control genes. Histograms represent the long-term coding adaptation values in 100 control sets. Red lines represent the long-term coding adaptation value in disease genes. Blue lines represent the long-term adaptation value across the whole coding genome. (A to E) All disease genes compared to controls. (F to J) Disease genes with many disease variants vs. controls, in low recombination regions of the genome. (A and F) Total alpha from ABC-MK. (B and G) Alpha for weak adaptation according to ABC-MK. (C and H) Alpha for strong adaptation according to ABC-MK. (D and I) Total alpha according to GRAPES. (E and J) Omega_a, the ratio of the rate of advantageous amino acid changes over the rate of synonymous changes, according to GRAPES.

number of deleterious segregating variants at a given locus is likely to vary significantly over evolutionary time due to genetic drift. This explanation, where the sweep deficit is specifically due to segregating deleterious variants, is particularly plausible given our results so far. Indeed, even though more selectively constrained genes, including disease genes, may be arguably more prone to harbor deleterious segregating variants because they are more constrained, we have already gathered evidence that purifying selection alone does not explain the sweep deficit at disease genes; it reflects the amount of deleterious variants that were removed, not the amount of currently segregating ones. Furthermore, we always compare disease and control non-disease genes with matched purifying selection. Lastly, we have shown that genes with a high level of purifying selection (high proportion of conserved elements) do not have any sweep deficit (see above).

The process of interference between deleterious and advantageous variants has been mostly studied in haploid species (*Jain, 2019*; *Johnson and Barton, 2002*; *Peck, 1994*). In diploid species including humans, recessive deleterious mutations specifically have been shown to have the ability to slow down, or even stop the frequency increase of advantageous mutations that they are linked with (*Assaf et al., 2015*). Dominant variants do not have the same interfering ability, because they do not increase in frequency in linkage with advantageous variants as much as recessive deleterious do, before the latter can be 'seen' by purifying selection when enough homozygous individuals emerge in a population (*Assaf et al., 2015*). (*Uricchio et al., 2019*) also found evidence of decreased protein adaptation in the regions of the human genome with strong background selection and low recombination. The majority of disease variants at mendelian disease genes are recessive (*Amberger et al., 2019*; *Balick et al., 2015*). Thus, if segregating recessive deleterious mutations are more common at disease genes, starting with the known disease variants themselves, then their interference could in theory explain the sweep deficit that we observe. This is true even despite the fact that we matched disease and control non-disease genes for multiple measures of selective constraint. Indeed, we use measures of selective constraint such as the density of conserved elements or the proportion of variable non-synonymous sites pN (Materials and methods), that are indicative of the amount of deleterious mutations that were ultimately removed, and not indicative of the number

of currently segregating deleterious variants. Disease genes and control non-disease genes may have very similar densities of conserved elements and similar pN, and still very different numbers of currently segregating recessive deleterious variants. Although directly comparing the actual total numbers of recessive deleterious mutations at disease and non-disease genes is difficult notably because estimating dominance coefficients in the human genome is a notoriously hard problem (*Huber et al., 2018*), we can still use indirect comparison strategies. First, if an interference of recessive deleterious mutations is involved then this interference is expected to be stronger in low recombination regions of the genome, where more deleterious mutations are likely to be genetically linked to an advantageous mutation. Therefore, we predict that the sweep deficit should be more pronounced when comparing disease and non-disease genes only in low recombination regions of the genome, where the linkage between deleterious and advantageous variants is higher. Conversely, the sweep deficit should be less pronounced in high recombination regions of the genome. Second, if the number of known segregating disease variants at a given disease gene correlates well enough with the total number of segregating recessive deleterious mutations at this disease gene then we should observe a stronger sweep deficit at disease genes with many known disease variants, compared to disease genes with few known segregating disease variants. Based on these two predictions, the sweep deficit should be particularly strong at disease genes with both many disease variants AND lower recombination. As the number of disease variants for each disease gene, we use the number of disease variants as curated by OMIM/UNIPROT (Materials and methods).

For these comparisons, we focus solely on African populations for which we found the strongest sweep deficit (*Figure 4*). We first compare disease and control non-disease genes both from only regions of the genome with recombination rates lower than the median recombination rate (1.137 cM/Mb). In agreement with recombination being involved, we find that the sweep deficit at low recombination disease genes is much more pronounced than the overall sweep deficit found when considering all disease and control non-disease genes regardless of recombination (*Figure 6*, FPR=$2.10^{-4}$). Conversely, the sweep deficit at disease genes compared to non-disease genes is much less pronounced when restricting the comparison to genes with recombination rates higher than the median recombination rate (1.137 cM/Mb), and remains only marginally significant (*Figure 6*, FPR=0.029). This provides evidence that genetic linkage may indeed be involved. Low

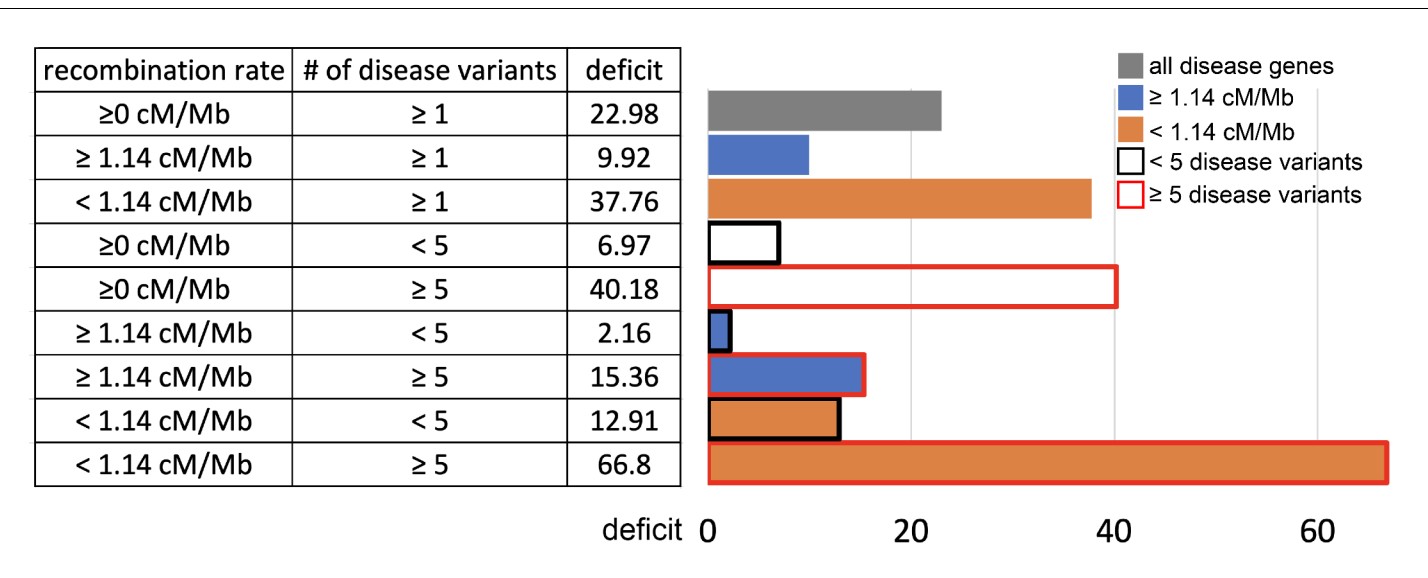

| recombination rate | # of disease variants | deficit |
|---|---|---|
| ≥0 cM/Mb | ≥ 1 | 22.98 |
| ≥ 1.14 cM/Mb | ≥ 1 | 9.92 |
| < 1.14 cM/Mb | ≥ 1 | 37.76 |
| ≥0 cM/Mb | < 5 | 6.97 |
| ≥0 cM/Mb | ≥ 5 | 40.18 |
| ≥ 1.14 cM/Mb | < 5 | 2.16 |
| ≥ 1.14 cM/Mb | ≥ 5 | 15.36 |
| < 1.14 cM/Mb | < 5 | 12.91 |
| < 1.14 cM/Mb | ≥ 5 | 66.8 |

Legend:
- all disease genes
- ≥ 1.14 cM/Mb
- < 1.14 cM/Mb
- < 5 disease variants
- ≥ 5 disease variants

**Figure 6.** Sweep deficit as a function of recombination and disease variants number. The sweep deficit is measured as the FPR score per gene (to make all tested groups comparable) over all window sizes, and $nS_L$ and iHS, as in *Figure 2* (Materials and methods). The different groups are separated according to recombination and numbers of disease variants so that they have approximately the same size (a half or a fourth of the disease genes). All deficits are measured using only African populations.

The online version of this article includes the following source data for figure 6:

**Source data 1.** Confounding factors differences between low and high recombination disease genes.

recombination is however not sufficient on its own to create a sweep deficit, and we further test if the sweep deficit also depends on the number of disease variants at each disease gene. In our dataset, approximately half of all the disease genes have five or more disease variants, and the other half have four or less disease variants (Materials and methods). In further agreement with possible interference of recessive deleterious variants, the sweep deficit is much more pronounced at disease genes with five or more disease variants (*Figure 6*, FPR=8.10$^{-4}$). The sweep deficit at disease genes with four or less disease variants is barely significant compared to control non-disease genes (*Figure 6*, FPR=0.032). In addition, disease genes with five or more disease variants, but with recombination higher than the median recombination rate, do not have a strong sweep deficit either (*Figure 6*, FPR=0.026). A higher number of disease variants alone is thus not enough to explain the sweep deficit. In a similar vein, disease genes with a recombination rate less than the median recombination rate, and with four or less disease variants, do not exhibit a strong sweep deficit (*Figure 6*, FPR=0.021). This confirms that low recombination alone is not enough to explain the sweep deficit at disease genes. Accordingly, disease genes with both low recombination AND five or more disease variants show the strongest sweep deficit (*Figure 6*, FPR=2.10$^{-4}$). Disease genes with both high recombination AND less than five disease variants show no sweep deficit at all, with a sweep prevalence undistinguishable from control non-disease genes (*Figure 6*, FPR=0.74). The latter result is important, because it suggests that interference of recessive deleterious variants may be sufficient on its own to explain the whole sweep deficit at disease genes. Both higher linkage and more disease variants seem to be needed to explain the sweep deficit at disease genes. Note that these results are not due to introducing a bias in the overall number of variants by using the number of disease variants, because we always match the level of neutral genetic variation between disease genes and control non-disease genes with pS. The overall level of genetic variation is further matched thanks to pN and thanks to McVicker's B, whose value is directly dependent on the level of genetic variation at a given locus (*McVicker et al., 2009*). Further note that only moderate differences in confounding factors between low and high recombination mendelian disease genes are unlikely to explain the sweep deficit difference (*Figure 6—source data 1*).

These results further imply that the alternative explanation of constitutively less common adaptation at disease genes is less likely than interference. With constitutively less adaptation at disease genes, the stronger sweep deficit observed at disease genes in low, compared to high recombination regions, could only reflect the fact that there is more statistical power to detect sweeps in low recombination regions (*Booker et al., 2020*; *O'Reilly et al., 2008*), and therefore more statistical power to distinguish a sweep deficit. A higher statistical power to detect sweeps in low recombination regions however does not explain the very strong sweep deficit in low recombination regions with many disease variants, and the marginal sweep deficit in low recombination regions with few disease variants.

We further find that the difference in sweep deficits between high and low recombination regions is not affected when using only nSL as a sweep statistic (Materials and methods). The nSL statistic was initially designed to be more robust to recombination than iHS (*Ferrer-Admetlla et al., 2014*), and to have more similar power in low and high recombination regions, and here we confirm this greater robustness. The two distributions of nSL sweep ranks, one for the lower recombination half and one for the higher recombination half of the genes, are much more similar than the two corresponding distributions of iHS sweep ranks (*Figure 7A,B*). Low recombination regions only have a slight excess of top-ranking nSL signals compared to high recombination regions. Such a small difference is unlikely to generate the substantial discrepancy in power needed to explain the much stronger sweep deficit in low recombination regions. The sweep deficit is substantial when using only nSL on all the disease genes and their controls (*Figure 7C*; FPR<5.10$^{-4}$). The nSL-only sweep deficit is only marginally significant in high recombination regions (FPR=0.043, deficit score=-9,227.4), but strongly significant and about four times more pronounced in low recombination regions (FPR<5.10$^{-4}$, deficit score=-33,177.2), the same relative difference observed when using both iHS and nSL (*Figure 6*).

More importantly, the fact that constitutively less adaptation at disease genes combined to more power to detect sweeps in low recombination regions does not explain our results, is made even clearer by the fact that disease genes in low recombination regions and with many disease variants have in fact experienced more, not less long-term adaptation according to an MK analysis using both ABC-MK and GRAPES (*Figure 5F,G,H,I,J*). ABC-MK in particular finds that there is a significant

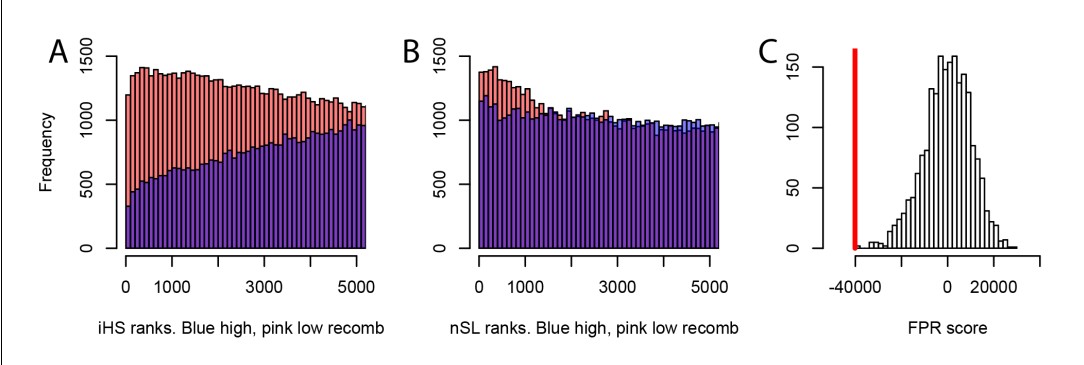

**Figure 7.** Different sweep detection power response of iHS and nSL to varying recombination rates. (**A**) iHS sweep ranks, shown from 1 to 5000 across all window sizes (50 kb to 1000 kb) in Africa, in low recombination (pink) or high recombination regions (blue). (**B**) Same as A. but for nSL. (**C**) Observed sweep deficit at disease genes (red line) compared to the distribution of the sweep deficit in 2000 block-randomized genomes. Same as *Figure 2A* but with only nSL.

excess of long-term strong adaptation (*Figure 5H*, P<0.01) in disease genes with low recombination and with many disease variants, compared to controls, but similar amounts of weak adaptation (*Figure 5G*, P=0.16). It might be that disease genes with many disease variants are genes with more mutations with stronger effects that can generate stronger positive selection. The potentially higher supply of strongly advantageous variants at these disease genes makes it all the more notable that they have a very strong sweep deficit in recent evolutionary times. This further strengthens the evidence in favor of interference during recent human adaptation: the limiting factor does not seem to be the supply of strongly advantageous variants, but instead the ability of these variants to have generated sweeps recently by rising fast enough in frequency.

## Decreased interference of recessive deleterious mutations during a bottleneck may explain the weaker sweep deficit in East Asia and Europe

An important observation in our analysis, that any potential explanation needs to account for, is the much weaker sweep deficit at disease genes in Europe and especially in East Asia, compared to Africa. If interference of recessive deleterious variants explains the sweep deficit at disease genes then it should also account for the weaker sweep deficit out of Africa. Previous results suggest that it might be the case. (*Balick et al., 2015*) showed that during a bottleneck of the magnitude of the Out of Africa bottleneck, there should be a sharp decrease of the segregating recessive deleterious variants load, because of all the low-frequency recessive deleterious variants that are removed when the bottleneck occurs. This is especially true for strongly deleterious variants that tend to segregate at lower frequencies. The magnitude of the bottleneck investigated by Balick et al. (a 10-fold decrease in population size) has since been confirmed for the Out of Africa bottleneck by the most recent Ancestral Recombination Graph approaches (*Speidel et al., 2019*). Populations in East Asia in particular went from an ancestral effective population size of ~10,000 to a post-Out of Africa effective population size of ~1000 for extended amounts of time before the very recent explosive human population expansions (*Speidel et al., 2019*). Balick et al. also found (i) evidence of an overall increased burden of recessive deleterious variants at disease genes compared to other genes and (ii) also found that this recessive burden had decreased in Europe, following the bottleneck out of Africa.

Here, we hypothesize that the bottleneck out of Africa decreased the recessive burden enough to cause a possible decrease of interference of recessive segregating variants at mendelian disease genes, and that this decrease of interference might explain the smaller sweep deficit observed at disease genes in Europe and especially in East Asia (*Figure 4*). We test this hypothesis using forward population simulations of loci with concentrations of deleterious variants meant to resemble a number of genic regions (Materials and methods). We find that, as expected given the results of Balick et al., there is much less interference of recessive deleterious variants after a bottleneck similar to

the Out of Africa bottleneck (*Table 1*). In *Table 1*, we provide both the fixation probabilities and the time to fixations of advantageous mutations of different strengths, under different simulated demographies matching either past demography in or out of Africa, and including deleterious mutations or not for comparing fixation parameters with or without interference (Materials and methods). In the presence of recessive deleterious variants, the time to fixation of advantageous variants in particular is only slightly increased after an Out of Africa-like bottleneck, compared to the strong fixation time increase when no bottleneck has taken place (*Table 1*). This interference effect is specific to recessive deleterious mutations and not observed with dominant deleterious mutations, as expected (*Assaf et al., 2015*). The effect on fixation time alone is likely sufficient to explain the sharp difference in sweep deficit observed, especially when comparing Africa and East Asia, the latter being the most bottlenecked population investigated here. Indeed, a sharp increase in fixation time is expected to result in substantially weaker sweep signals. This is because a slower increase in frequency of an advantageous mutation will leave more time to a larger number of recombination events to occur, and thus narrow down the breadth of the sweep signature around that advantageous mutation (*Assaf et al., 2015*). A reduction of the segregating recessive burden as observed by Balick et al. at mendelian disease genes, and therefore interference, can thus explain the

**Table 1.** Decreased interference during a bottleneck.

The table provides the proportion of advantageous mutations that go to fixation (% fixed), and the time to fixation under multiple conditions simulated with SLiM (Materials and methods). For example, s=0.005, 40% constrained, recessive means that we simulate advantageous mutations with s=0.005, surrounded by a genomic region where 40% of sites experience recessive deleterious mutations according to a specific distribution of fitness effects (Materials and methods). The fix. time increase column provides the relative increase in fixation time (ratio of times) in the presence compared to in the absence of deleterious mutations. The time to fixation is in number of generations. The Methods provide more details on the simulations.

| | Demography | Deleterious mutations? | Time to fixation | % fixed | Fix. time increase | Fix. prob decrease |
|---|---|---|---|---|---|---|
| **S=0.005, 10% constrained, recessive** | **East Asia: 10000->1000** | **No** | **2265** | **0.0050** | | |
| | | **Yes** | **2547** | **0.0033** | **1.12** | **0.66** |
| | **Africa: 10000->10000** | **No** | **4204** | **0.0051** | | |
| | | **Yes** | **6530** | **0.0038** | **1.55** | **0.74** |
| s=0.005, 20% constrained, recessive | demography | deleterious mutations? | time to fixation | % fixed | fix. time increase | fix. prob decrease |
| | East Asia: 10000->1000 | no | 2265 | 0.0050 | 1.16 | 0.66 |
| | | yes | 2617 | 0.0033 | | |
| | Africa: 10000->10000 | no | 4204 | 0.0051 | 1.69 | 0.62 |
| | | yes | 7113 | 0.0032 | | |
| s=0.005, 40% constrained, recessive | demography | deleterious mutations? | time to fixation | % fixed | fix. time increase | fix. prob decrease |
| | East Asia: 10000->1000 | no | 2265 | 0.0050 | 1.17 | 0.65 |
| | | yes | 2642 | 0.0033 | | |
| | Africa: 10000->10000 | no | 4204 | 0.0051 | 2.10 | 0.56 |
| | | yes | 8809 | 0.0028 | | |
| s=0.01, 40% constrained, recessive | demography | deleterious mutations? | time to fixation | % fixed | fix. time increase | fix. prob decrease |
| | East Asia: 10000->1000 | no | 1530 | 0.0092 | 1.37 | 0.78 |
| | | yes | 2090 | 0.0072 | | |
| | Africa: 10000->10000 | no | 2546 | 0.0098 | 2.05 | 0.82 |
| | | yes | 5209 | 0.0080 | | |
| s=0.005, 40% constrained, dominant | demography | deleterious mutations? | time to fixation | % fixed | fix. time increase | fix. prob decrease |
| | East Asia: 10000->1000 | no | 2265 | 0.0050 | 0.96 | 0.93 |
| | | yes | 2169 | 0.0046 | | |
| | Africa: 10000->10000 | no | 4204 | 0.0051 | 0.99 | 0.92 |
| | | yes | 4169 | 0.0047 | | |

observed patterns in Africa versus East Asia and Europe. This reduction might be due to a number of concurrent processes: a reduction in the number of recessive segregating variants as observed by Balick et al., but hypothetically, might also be due to a decrease in the effective deleteriousness of the remaining segregating variants due to the lower effective population size. This further supports the idea that interference with recessive deleterious variants may explain our observation of a strong sweep deficit at disease genes in Africa, and of weaker sweep deficits out of Africa.

### Similar levels of sweep depletion in mendelian disease genes across MeSH disease classes

Because we find an overall sweep depletion at mendelian disease genes, we further ask if genes associated with different diseases might show different patterns of depletion (always in African populations). We classify disease genes into different classes according to the Medical Subject Headings (MeSH) annotation for diseases in DisGeNet (*Piñero et al., 2020*). The MeSH annotations organize the disease genes into broad disease categories that overlap with distinct organs or large physiological systems (for example the endocrine system). We find significant (FPR<0.05) sweep depletions in Africa for all but one disease MeSH classes (FPR<0.05; *Figure 8*). The sweep deficit is comparable across MeSH disease classes (*Figure 8*), suggesting that the evolutionary process at the origin of the sweep deficit is not disease-specific. This is compatible with a non-disease specific explanation such as recessive deleterious variants interfering with adaptive variants, irrespective of the specific disease type. The only non-significant deficit is for the MeSH term immune system diseases. Interestingly, there is evidence that past adaptation at disease genes in response to diverse pathogens has resulted in increased prevalence of specific auto-immune diseases (*Barreiro and Quintana-Murci, 2010*), and we can speculate that this might be why we do not see a sweep deficit at those genes.

## Discussion

We found a depletion of the number of genes in recent sweeps at human non-infectious, mendelian disease genes compared to non-disease genes. Although more work is needed, the lack of sweeps

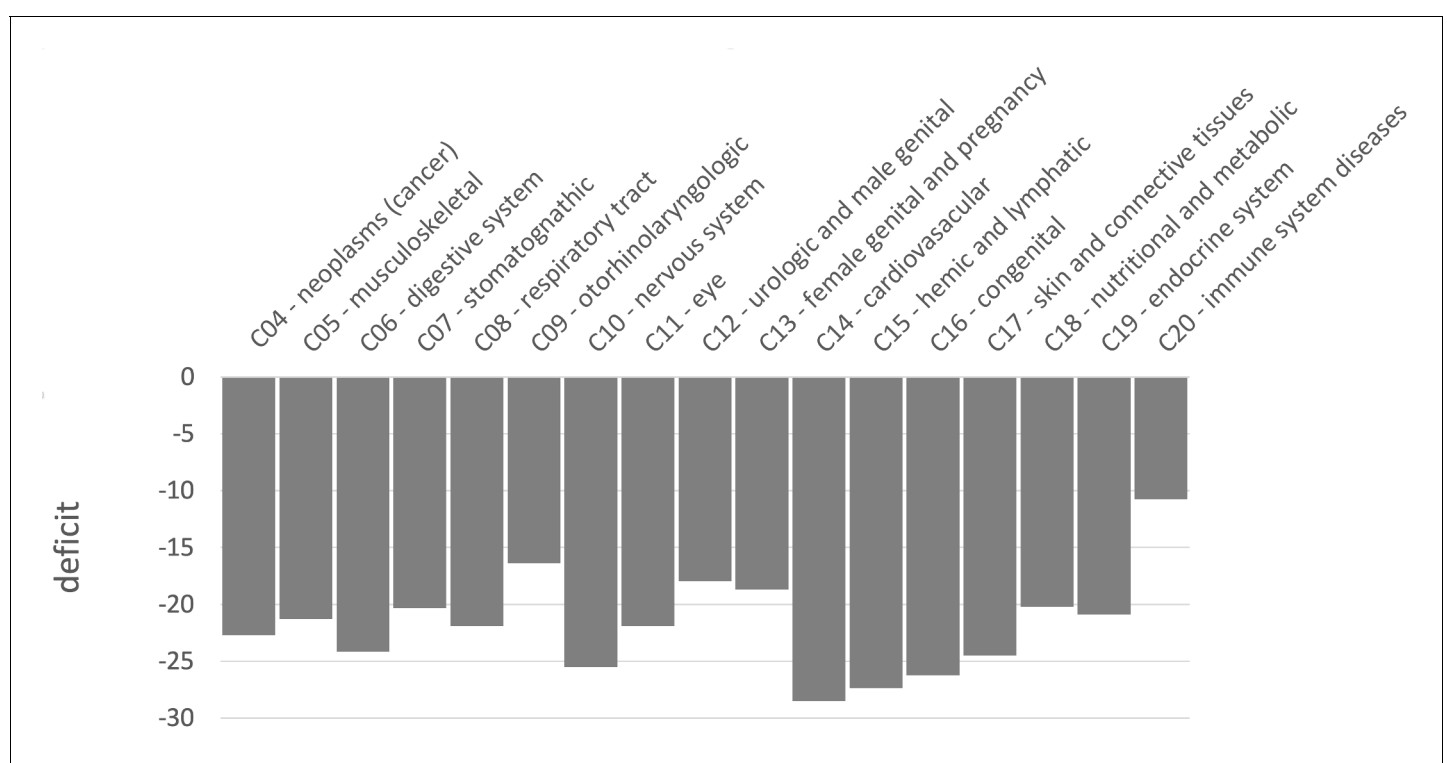

**Figure 8.** Sweep deficit per MeSH disease classes. The sweep deficit is measured as the overall FPR score per gene (Materials and methods), to make all MeSH classes comparable even if they include different numbers of genes.

at disease genes already favors specific evolutionary processes over others. For example, it makes it unlikely that past adaptations increasing the occurrence of disease variants through hitchhiking would be the dominant process linking disease and adaptation at the gene level. The lack of sweeps at mendelian disease genes especially in Africa also seems to be unrelated to any difference in mutation accumulation between disease and non-disease genes, since we find no sign of a difference in mutation rates between the two categories of genes in the first place, and since we match metrics accounting for mutation rate in our comparisons (e.g., GC content and pS). Instead, a lack of sweeps, once selective constraint has been controlled for, seems to favor a relationship involving a decrease of recent adaptation at disease genes, beyond simple constraint (measured by the amount and strength of deleterious mutations that are removed).

Multiple mechanisms might explain such a lack of recent adaptation. A first possible hypothesis is that disease genes are genes that can be sensitive to the environment and whose fitness optimum can change during evolution when the environment changes. However, when this happens, adaptation may be constitutively less common at disease genes. Although higher pleiotropy is a tempting hypothesis to explain such a lag (*Otto, 2004*), disease genes have not experienced less long-term protein adaptation. Since gene pleiotropy is a stable property over evolutionary time, it is difficult to see how it would generate a recent adaptation deficit without also generation a deficit of long-term adaptation. This also likely excludes that mendelian disease genes just happen to be genes where mutations are rarely advantageous because they have strong effects that tend to overshoot, and thus miss the fitness optimum. On the contrary, we find that mendelian disease genes (and their controls) have experienced more long-term protein adaptation than the genomic baseline (*Figure 5*). Given our results on genetic interference, disease genes may experience as much, or even more long-term adaptive substitutions while also showing a deficit of strong recent adaptation because the list of current human disease genes defined by the presence of segregating disease variants varies over evolutionary time. The list of genes that are mendelian disease genes may evolve over evolutionary time based on where the transiently segregating, recessive deleterious variants that define them, are found in the genome as a result of the interplay between gene constraint and genetic drift. In this case, the millions of years of human evolution would be more than enough to see a substantial turnover of genes with segregating, recessive deleterious variants in the genome.

A plausible explanation for all our observations is indeed genetic interference, where selective sweeps are impeded at disease genes due to the interference of genetically linked recessive deleterious variants. The deleterious effects of these variants can reveal themselves when they hitchhike together with an advantageous variant that is just starting to increase in frequency (*Assaf et al., 2015*). Accordingly, we find a marked sweep depletion in Africa when restricting the comparison to disease and non-disease genes in low recombination regions of the genome and with higher numbers of disease variants (*Figure 6*). We also show through simulations that a stronger sweep deficit in Africa is expected if genetic interference indeed explains our results. All these comparisons are however indirect; we do not quantify directly the effect of recessive deleterious mutations at disease or non-disease genes. That said, the majority of mendelian disease variants are known to be recessive (*Balick et al., 2015*), and using the number of disease variants, as done in the present study, should be a good proxy of the actual number of segregating recessive deleterious mutations. Estimating dominance may prove challenging, however, since it is difficult to distinguish selection coefficient changes from dominance coefficient changes (*Huber et al., 2018*). Again, our results provide preliminary evidence to further test in the future.

In addition to suggesting possible explanatory evolutionary scenarios, our results highlight a number of potential limitations and biases that also need to be further explored. First, the lack of sweeps at disease genes suggests the possibility of a technical bias against the annotation of disease genes in sweep regions with high LD, as described in the Results. This bias is unlikely to be the dominant explanation for our results, because then we would expect a stronger sweep deficit at disease genes in Europe than in Africa, given that most disease genes were annotated in Europe. The recombination rate at disease genes is also only slightly different from the recombination rate at non-disease genes (*Figure 1*), and we match the recombination rate between disease genes and controls. The increase of the sweep deficit when comparing disease and non-disease genes only in low recombination regions (*Figure 6*), where disease annotation would then be more difficult regardless of overlapping a sweep or not, also suggests that this bias is unlikely.

Further work is now required regarding the connection between the sweep deficit and polygenic adaptation not leaving hitchhiking signals. Our results could also be explained by a different balance between sweeps and polygenic adaptation at mendelian disease genes, with less sweeps but more polygenic adaptation that would be less affected by interference with deleterious variants. That said, we do find that mendelian disease genes have experienced more long-term adaptive protein evolution than the genomic baseline (*Figure 5*), suggesting mutations that were advantageous enough to go all the way to fixation. It may be possible to use recent polygenic adaptation quantification tools such as PALM (*Stern et al., 2021*) to compare its prevalence between mendelian disease and non-disease genes.

Finally, there are multiple directions to further analyze the sweep deficit at disease genes that we have not explored in this manuscript. For instance, analyzing the sweep deficit as a function of the time of onset of diseases (early or late in life), might further provide clues to why the sweep deficit exists in the first place. Preliminary comparison of the sweep deficit at specific MeSH disease classes (*Figure 8*) with known early (congenital diseases) or mostly late onsets (cancer, cardiovascular), however, suggests that the average onset time of diseases might not make much of a difference.

In conclusion, although our analysis reveals a strong deficit of selective sweeps at human disease genes in Africa that seems to be due to genetic interference, it also suggests that more work is needed to better understand the evolutionary processes at work, and the biases that may have skewed our interpretations. Despite these limitations, our comparison already suggests that specific evolutionary relationships between disease genes and adaptation might be more prevalent than others, especially interference between segregating recessive deleterious and advantageous variants. As an important follow-up question, it may now be important to ask how the sweep deficit at disease genes might have hidden interesting adaptive patterns in previous functional enrichment analyses, especially in gene functions that are often annotated based on disease evidence in the first place. For example, metabolic genes are believed to be of particular interest for adaptation to climate change. But metabolic genes are often found due to their role in metabolic disorders, and a strong representation of disease genes among all metabolic genes could then in theory mask any sweep enrichment. A sweep enrichment at metabolic genes might only become visible once controlling for the proportion of disease genes, in addition to the list of controls that we already use in the present analysis. Our results thus highlight the great complexity of studying functional patterns of adaptation in the human genome.

## Materials and methods

### Disease gene lists

We consider genes that are known to be associated (mendelian type of association) with diseases as mendelian disease genes. We focus on protein-coding genes associated with human mendelian non-infectious diseases. By non-infectious, we mean that we excluded genes with known infectious disease-associated variants. This does not exclude most virus-interacting genes since most of them are not associated at the genetic variant level with infectious diseases. It is important to note that the effect of virus interactions is accounted for by matching the number of interacting viruses between mendelian disease genes and controls (see below). Complex diseases are associated with several loci and environmental factors. Patterns of positive selection at complex disease and mendelian disease genes may differ (*Blekhman et al., 2008*; *Quintana-Murci, 2016*; *Torgerson et al., 2009*), which is why we restrict our analysis to mendelian disease genes. We also restrict our analyses to non-infectious disease genes, since disease, genetic associations with pathogens are an entirely different problem. We nevertheless control for the proportion of genes that are immune genes or interact with viruses (see below), since it has been shown that immune genes and interactions with viruses drive a large proportion of genomic adaptation in humans (*Enard and Petrov, 2020*). Therefore, different proportions of immune and virus-interacting genes between disease and non-disease genes might confound their comparison. Moreover, although diseases can be associated with non-coding genes, we only use protein-coding genes. We curate disease genes defined as genes associated with diseases in a mendelian fashion according to both DisGeNet (*Piñero et al., 2020*) and OMIM (*Amberger et al., 2019*), to ensure that we focus on high-confidence mendelian disease genes. DisGeNet is a comprehensive database including gene-disease associations (GDAs) from

many sources. In order to get disease genes with high confidence, we further only use GDAs curated by UniProt. These gene-disease associations are extracted and carefully curated from the scientific literature and the OMIM (Online Mendelian Inheritance in Man) database, which reports phenotypes either mendelian or possibly mendelian (*Amberger et al., 2019*). We also exclude all genes associated with infectious diseases according to MeSH annotation (disease class C01). In the end, we curate 4215 non-infectious mendelian disease genes from DisGeNet also curated by OMIM and Uniprot. Although we rely on GDAs from Uniprot to curate high-quality disease genes, we also include GDAs of DisGeNet from other sources when classifying disease genes into different MeSH classes and measuring pleiotropy, as long as a disease gene has at least one GDA curated by OMIM and Uniprot. We completely exclude GDAs that are only reported by CTD (Comparative Toxicogenomics Database) (*Davis et al., 2021*) in this study. This is because CTD includes a broad range of chemical-induced diseases that might only happen when people are exposed to these chemicals, especially some inorganic chemicals that may not be present in natural environments (*Davis et al., 2021*).

In order to study different types of diseases, we also divide disease genes into different classes according to the annotated MeSH classes in DisGeNet (*Piñero et al., 2020*). Those diseases without MeSH class are annotated as 'unclassfied'. Genes belonging to more than one MeSH class are counted in each MeSH class where they are present. MeSH classes including less than 50 genes are not considered in this study. We classify all the non-infectious disease genes into 17 MeSH classes including Neoplasms (C04), Musculoskeletal Diseases (C05), Digestive System Diseases (C06), Stomatognathic Diseases (C07), Respiratory Tract Diseases (C08), Otorhinolaryngologic Diseases (C09), Nervous System Diseases (C10), Eye Diseases (C11), Male Urogenital Disease (C12), Female Urogenital Diseases and Pregnancy Complications (C13), Cardiovascular Diseases (C14), Hemic and Lymphatic (C15), Congenital, Hereditary, and Neonatal Diseases and Abnormalities (C16), Skin and Connective Tissue Diseases (C17), Nutritional and Metabolic Diseases (C18), Endocrine System Diseases (C19), Immune System Diseases (C20), and 'unclassified'.

## Detecting recent selection signals at human genes

All the analyses were conducted human genome version hg19. We use two different methods to detect selective sweeps in human populations: iHS (*Voight et al., 2006*) and nSL (*Ferrer-Admetlla et al., 2014*). Both approaches are haplotype-based statistics calculated with polymorphism data. We use human genome data from the 1,000 Genomes Project phase 3, which includes 2504 individuals from 26 populations (*Auton et al., 2015*).

We measure iHS and nSL in windows centered on human coding genes (i.e. windows whose center is located half-way between the most upstream transcript start site and most downstream transcript stop site of protein coding genes). We use windows of sizes ranging from 50 kb to 1000 kb (50kb, 100kb, 200kb, 500kb and 1000kb) since we do not want to presuppose of the size of sweeps, and since the size of the selective sweeps may vary between different genes. Moreover, to avoid any preconception related to the expected strength or number of sweep signals, we use a moving rank threshold strategy to measure the enrichment or deficit in sweeps at disease genes. For example, we select the top 500 genes with the stronger sweep signals according to a specific statistic (iHS or nSL). We then compare the number of diseases and non-disease genes within the top 500 genes with the strongest iHS or $nS_L$ signals. This was repeated for different top thresholds and the corresponding ranks from top 5,000 to top 10 (5000,4000,3000,2500,2000,1500,1000,900,800,700,600,500,450,400,350,300,250,200,150,100,90,80,70,60,50,40,30,25,20,15,10). Using a range of rank thresholds makes less assumptions and provides more flexibility than the classic outlier approach, even though we still have to arbitrarily determine a list of rank thresholds to include. This is because we can get a significant result not only due to an enrichment of only the top, absolutely strongest sweeps, but also due for example to a large excess of weak or moderate sweeps, that would for example increase the expected numbers in the top 5000 or top 2000, without increasing the number of sweeps in the top 100 or top 50. Therefore, our approach is sensitive to a more diverse range of sweeps than the classic outlier approach, that makes a very restrictive assumption that sweeps have to be necessarily be strong. Genes are ranked based on the average iHS or nSL in their gene centered windows. Both iHS and nSL measure, individually for each SNP in the genome, how much larger haplotypes linked to the derived SNP allele are compared to haplotypes linked to the ancestral allele. For each window, we measure the average of the absolute value of iHS or $nS_L$ over all the SNPs in that window with an

iHS or $nS_L$ value. The average iHS or nSL values in a window provide high power to detect recent select sweeps (**Enard and Petrov, 2020**).

## Comparing recent adaptation between disease and non-disease genes

We use a previously developed gene-set enrichment analysis pipeline to compare recent adaptation between disease and non-disease genes (**Enard and Petrov, 2020**) available at https://github.com/ DavidPierreEnard/Gene_Set_Enrichment_Pipeline. This pipeline includes two parts. The first part is a bootstrap test that estimates the whole sweep enrichment or depletion curve at genes of interest (mendelian disease genes in our case) while controlling for confounding factors. The second part is a false positive risk (also known as false discovery rate in the context of multiple testing) that estimates the statistical significance of the whole sweep enrichment curve using block-randomized genomes (**Enard and Petrov, 2020**).

To compare disease and non-disease genes, we first need to select control non-disease genes that are sufficiently far away from disease genes. In that way, we avoid using as controls non-disease genes that overlap the same sweeps as neighboring disease genes, thus resulting in an underpowered comparison. The question is then how far do we need to choose non-disease control genes? Ideally, we would choose non-disease control genes as far as possible from disease genes in the human genome, further than the size of the largest known sweeps (e.g. the lactase sweep), which would be on the order of a megabase. However, because there are many disease genes in our dataset (4215), there are very few non-disease genes in the human genome that are more than one megabase away from the closest disease gene. This is a problem, because the available number of potential control non-disease genes is an important parameter that can affect both the type I error, false positive rate, and type II error, false negative rate of the disease vs. non-disease genes comparison. Indeed, the smaller the control set, the more likely it is to deviate from being representative of the true null expectation at non-disease genes. The noise associated with a small sample could go either way. Either the small control sample happens by chance to have less sweeps, and the bootstrap test we use to compare disease and non-disease genes will become too liberal to detect sweep enrichments, and to conservative to detect sweep deficits. Or the small control sample happens by chance to have more sweeps than a larger control sample would, and the bootstrap test becomes too conservative to detect sweep enrichments, and too liberal to detect sweep deficits.

After trying distances between disease genes and control disease genes of 100 kb, 200 kb, 300 kb, 400 kb, and 500 kb, we find that the sweep deficit observed at disease genes increases steadily from 100 kb to 300 kb (*Table 2*), showing that 100 kb or 200 kb are likely insufficient distances. Further than 300 kb at 400 kb, we do not observe much stronger sweep deficits than at 300 kb, while at the same time the risks of type I and type II errors keep increasing due to shrinking non-disease genes control sets. This would translate in a decreased power to possibly exclude the null hypothesis of no sweep enrichment or deficit in the second part of the pipeline, when estimating the actual pipeline FPR. Because of this, we set the required distance of potential control non-disease genes from disease genes at 300 kb. This is also the distance where there are still approximately as many control genes (3455) as there are disease genes that we can use for the comparison (3030; those genes out of the 4215 disease genes with sweep data and data for all the confounding factors).

Another important aspect of the bootstrap test (first part of the pipeline), aside from setting up the minimal distance of the control non-disease genes, is the matching of potential confounding

**Table 2.** Sweep deficit as a function of the minimal distance of control non-disease genes.
The sweep deficit is measured by the FPR score, that is the cumulative difference between the number of genes in sweeps at disease and control non-disease genes, across window sizes, sweep summary statistics, and African populations (see the rest of the Materials and methods).

| Minimal distance | Sweep deficit |
| --- | --- |
| 100 kb | −20889 |
| 200 kb | −35009 |
| 300 kb | −68928 |
| 400 kb | −88546 |

factors likely to influence sweep occurrence. We choose non-disease control genes that have the same confounding factors characteristics as disease genes (for example, control non-disease genes that have the same gene expression level across tissues as disease genes). The precise matching algorithm is detailed in *Enard and Petrov, 2020*. In brief, the bootstrap test builds sets of control genes that have the same overall average values for confounding factors as disease genes. For example, the bootstrap test can build 100 control sets, with each set having the same overall average GC content as disease genes. Note that this means that disease genes are not individually matched one by one with one control gene that happens to have the same GC content. Matching genes individually, instead of matching the overall gene sets averages, would indeed limit the pool of potential control genes too drastically. For more details on this, please refer to *Enard and Petrov, 2020*.

When comparing disease and non-disease genes with the bootstrap test, we control for the following potential confounding factors that could influence the occurrence of sweeps at genes:

- Average overall expression in 53 GTEx v7 tissues (*GTEx Consortium, 2020*) (https://www.gtexportal.org/home/). We used the log (in base 2) of TPM (Transcripts Per Million).
- Expression (log base 2 of TPM) in GTEx lymphocytes. Expression in immune tissues may impact the rate of sweeps.
- Expression (log base 2 of TPM) in GTEx testis. Expression in testis might also impact the rate of sweeps.
- deCode recombination rates 50 kb and 500 kb: recombination is expected to have a strong impact on iHS and nSL values, with larger, easier to detect sweeps in low recombination regions but also more false positive sweeps signals. The average recombination rates in the gene-centered windows are calculated using the most recent deCode recombination map (*Halldorsson et al., 2019*). We use both 50 kb and 500 kb window estimates to account for the effect of varying window sizes on the estimation of this confounding factor (same logic for other factors where we also use both 50 kb and 500 kb windows).
- GC content is calculated as a percentage per window in 50 kb and 500 kb windows. It is obtained from the USCS Genome Browser.
- The density of coding sequences in 50 kb and 500 kb windows centered on genes. The density is calculated as the proportion of coding bases respect to the whole length of the window. Coding sequences are Ensembl v99 coding sequences.
- The density of mammalian phastCons conserved elements (*Siepel et al., 2005*) (in 50 kb and 500 k windows), downloaded from the UCSC Genome Browser. We used a threshold considering 10% of the genome as conserved, as it is unlikely that more than 10% of the whole genome is constrained according to previous evidence (*Siepel et al., 2005*). Given that each conserved segment had a score, we considered those segments above the 10% threshold as conserved.
- The density of regulatory elements, as measured by the density of DNASE1 hypersensitive sites (in 50 kb and 500 kb windows) also from the UCSC Genome Browser.
- The number of protein-protein interactions (PPIs) in the human protein interaction network (*Luisi et al., 2015*). The number of PPIs has been shown to influence the rate of sweeps (*Luisi et al., 2015*). We use the log (base 2) of the number of PPIs.
- The gene genomic length, that is the distance between the most upstream and the most downstream transcription start sites.
- The number of gene neighbors in a 50 kb window, and the same number in 500 kb window centered on the focal genes: it is the number of coding genes within 25 kb or within 250 kb.
- The number of viruses that interact with a specific gene (*Enard and Petrov, 2020*).
- The proportion of immune genes. The matched control sets have the same proportion of immune genes as disease genes, immune genes being genes annotated with the Gene Ontology terms GO:0002376 (immune system process), GO:0006952 (defense response) and/or GO:0006955 (immune response) as of May 2020 (*Gene Ontology Consortium and Gene Ontology, 2021*).
- The average non-synonymous polymorphism rate pN in African populations, and the the synonymous rate pS. We matched pN to build control sets of non-disease genes with the same average amount of strong purifying selection as disease genes. Also, pS can be a proxy for mutation rate and we can build control sets of non-disease genes with similar level of mutation rates.
- McVicker's B value which can be used to account for more recent selective constraint (*McVicker et al., 2009*).

Similar to the selection of control genes far enough from disease genes, the matching of many confounding factors decreases the number of non-disease genes that can effectively be used as controls. This further increases the risk of type I and type II errors of the bootstrap test, as previously described (*Enard and Petrov, 2020*). In addition, the bootstrap test only provides a p-value for each tested sweep rank threshold separately, in the whole enrichment (or deficit) curve (*Figure 3*). It does not provide any estimate of the significance of the whole curve, which is needed to estimate the significance of a sweep enrichment or deficit without making too many assumptions on how many sweeps are expected or how strong they are.

To address the increased type I and type II error risks of the bootstrap test, as well to get an unbiased significance estimate for whole enrichment curves, the second part of our pipeline conducts a false positive risk (FPR) analysis based on block-randomized genomes (*Enard and Petrov, 2020*). Briefly, we re-estimate many whole enrichment curves reusing the same mendelian disease and control non-disease genes used in the first part of the pipeline by the bootstrap test, but after having randomly shuffled the locations of genes or clusters of neighboring genes in sweeps at those disease and control non-disease genes. To do this, we order the disease and control non-disease genes as they appear in the genome. We then define blocks of neighboring genes, whose limits do not interrupt clusters of genes in the same putative sweep. Then, we randomly shuffle the order of these blocks. Because we do not cut any cluster of genes that might be in the same sweep, the resulting block-randomized genomes preserve the same clustering of the genes in the same putative sweeps as in the real genome. With this approach, we look at the exact same set of disease and control non-disease genes and just shuffle sweep locations between them. Thus, by using many block-randomized genomes, we can estimate the null expected range of whole enrichment curves while fully accounting for the extra variance expected from having a limited sample of control non-disease genes. We can then estimate a false positive risk (FPR) for the whole enrichment or deficit curve by comparing the real observed one with the distribution of random curves generated with block-randomized genomes.

To measure the FPR for a curve, we need to define a metric to compare the real curve with the randomly generated ones. In *Figure 3*, we show relative enrichments at each sweep rank threshold, the number of disease genes in sweeps divided by the number of control non-disease genes in sweeps. As a summary metric for the curve, we could then use the sum of the relative enrichments over all thresholds. However, the issue with this approach is that a relative enrichment is the same whether we have two disease genes in sweeps and one control non-disease gene in sweeps, or we have 200 disease genes in sweeps and 100 control non-disease genes in sweeps. Thus, although relative enrichments are convenient for visualization on a figure, they are not adequate to measure the FPR. Instead of the relative enrichment, we use as a score for estimating the FPR (FPR score) the difference between disease and non-disease genes, that is, the number of disease genes in sweeps, minus the average number of control non-disease genes across control sets built by the bootstrap test. We then get this score for a whole curve the sum of differences over all the rank thresholds. We use this sum of differences to estimate the enrichment or deficit curve FPR, as the proportion of block-randomized genomes where the FPR score (the sum of differences) exceeds the observed sum of differences for an enrichment (one minus this proportion for a deficit).

We can write this FPR score as follows. With t being the number t threshold belonging to T, the set of rank threshold numbers, $D_t$ the number of disease genes in sweeps at threshold number t, and $C_t$ the number of control genes in sweeps at threshold number t then:

$$\mathrm{FPRscore} = \sum_{t \in T}(D_t - C_t)$$

Importantly, although so far we have described the case where we measure the FPR for one enrichment curve, nothing prevents us from calculating a single sum of differences over an entire group of enrichment or deficit curves. This way, we can measure a single FPR for any number of curves considered together. In our analysis, we measure a single FPR adding iHS and $nS_L$ curves together, and also adding together the curves for 50kb, 100kb, 200kb, 500kb, and 1000kb windows (10 curves in total, 2 statistics*5 window sizes). Then, with W the set of window sizes, M the set of summary statistics used for detecting sweeps, and P the set of populations, we have

$$\mathrm{FPRscore} = \sum_{m \in M} \sum_{w \in W} \sum_{p \in P} \sum_{t \in T} \left( D_{t,m,w,p} - C_{t,m,w,p} \right)$$

In this FPR score, it is important to note that the strongest sweeps signals in the top ranks weight more than the weaker ranks. For example, if the top rank threshold used is 10 for the top 10 genes with the strongest sweep signals, then a disease gene in the top 10 is also in the top 20, or top 50, or top 100 or any other less restrictive defined rank threshold. Such a disease gene thus contributes to $D_1$, $D_2$, $D_3$ (....) up to $D_n$, where n is the number of rank thresholds used. It follows that the genes in the top rank threshold are weighted by a factor of n, and that the genes in the second top rank threshold, but not in the top rank threshold, are weighted by a factor of n-1, and so on up to the genes in the last nth rank threshold being weighted by a factor of 1. We can thus define $d_t$ as the number of genes with ranks lower than rank threshold number t, but higher than rank threshold number t-1. Then, with $c_t$ being the equivalent of $d_t$ but for control genes:

$$\mathrm{FPRscore} = \sum_{m \in M} \sum_{w \in W} \sum_{p \in P} \sum_{t \in T} \left( d_{t,m,w,p} - c_{t,m,w,p} \right) * (n + 1 - t)$$

This weighting scheme is justified, as it makes sense to give more weight to stronger, and therefore higher confidence sweep signals.

## Additional GERP confounding factors

To test if the confounding factors enumerated above properly account for purifying selection/selective constraint when comparing disease and control genes, we add several GERP-based metrics (*Davydov et al., 2010*) to the list of matched confounding factors, and check whether it makes a difference or not when estimating the sweep deficit in disease genes. Indeed, GERP provides a quantification of purifying selection in a given genomic window. So if adding GERP data to the confounding factors makes a difference for the sweep deficit, then it means that purifying selection was not already accounted for by the existing confounding factors. On the contrary, if adding GERP data to the confounding factors makes no difference for the sweep deficit, then it shows that the already included confounding factors, already meant to account for purifying selection such as the density of phastCons conserved elements or McVicker's B statistic (among others), already control well for purifying selection. As additional confounding factors, we consider the average GERP score in 50 kb and 500 kb windows centered on genes, as well as the density of GERP conserved elements also in 50 kb and 500 kb windows (four additional confounding factors in total) downloaded from the Sidow lab website for human genome assembly hg19 (http://mendel.stanford.edu/SidowLab/downloads/gerp/).

## McDonald-Kreitman analysis of long-term coding adaptation

We use ABC-MK (https://github.com/jmurga/Analytical.jl, *Moreno, 2021*) and GRAPES (https://github.com/BioPP/grapes, *Dutheil, 2021*) to estimate the long-term rate of protein adaptation (both with the alpha and the omega_a estimates, see *Figure 5*). As divergence data to get DN (number of non-synonymous fixed substitutions) and DS (number of synonymous fixed substitutions), we count the number of human-specific fixed substitutions (*Uricchio et al., 2019*). As non-synonymous and synonymous genetic variation data, we use the variants from the 1000 Genomes phase 3 for the 661 African individuals included (*Uricchio et al., 2019*). We use GRAPES to estimate the average strength of deleterious non-synonymous variants with the DisplGamma distribution (*Galtier, 2016*).

## Sweep deficit at high and low recombination disease genes, and at high and low disease variant number disease genes

To generate *Figure 6*, we separate disease genes in groups of approximately the same size based on their recombination rate and numbers of disease variants annotated in OMIM/Uniprot. We separate the disease genes into two groups of equal size, those with recombination lower than 1.137 cM/Mb, and those with recombination higher than this value. To count the disease variants at each disease gene, we count not only the OMIM/Uniprot disease variants for that gene, but also all the other OMIM/Uniprot disease variants that occur in a 500 kb window centered on that gene. We do this because the recessive deleterious variants form other nearby disease genes may also interfere

with adaptation. Half of disease genes have less than five OMIM/Uniprot disease variants, and half have five or more.

## Selection of the 1000 genes with the highest density of conserved elements

To compare highly constrained genes with other genes in the genome to check if purifying selection alone can create a sweep deficit, we first identify the 1000 genes with both complete confounding factors and complete sweep data. We then ask if these 1000 genes have a sweep deficit compared to other genes in the genome far enough from them (>300kb as for the disease vs. control genes). For this comparison, we cannot match the following confounding factors because they are themselves measures of, or related to purifying selection: the density of conserved elements, coding and regulatory densities, pN, pS, the number of gene neighbors and gene's genomic length. We still match all other confounding factors.

## Population simulations of interference with and without a bottleneck

To investigate if reduced interference of recessive deleterious variants could explain the waker sweep deficit observed, we use SLiM (*Haller and Messer, 2019*) to simulate advantageous mutations in the presence of recessive deleterious mutations. To estimate the average fixation times and probabilities included in *Table 1*, we get the average of these values over 1000 simulations each time. We simulate a one megabase genomic region. Across this entire region, we set the percentage of conserved sites that can experience deleterious mutations at 8%, believed to be the average proportions of sites that are conserved in the human genome (*Davydov et al., 2010*; *Siepel et al., 2005*). At the center of the one megabase region, we include a 100 kb region where the proportion of sites that can experience deleterious mutations is higher, from 10%, to 20%, to 40%. This is meant to simulate the fact that we center our analysis on genes. The 10% proportion of sites that can experience deleterious mutations represents approximately the median GERP density of conserved elements in 100 kb windows centered on genes (*Figure 9*). The 20% and 40% proportions represent moderately elevated, and very strongly elevated values, respectively (*Figure 9*). We simulate a distribution of deleterious fitness effects (DFE) with a relatively flat profile across orders of magnitude for s, as found by recent human DFE estimates (*Kim et al., 2017*), with four negative selection coefficients, from weakly to strongly deleterious (s=-0.002,–0.02,−0.1,–0.5). The population size is initially set to 10,000 individuals and stays that way to simulate African populations (*Speidel et al., 2019*). After a burn-in period of 20,000 generations or 2N (deleterious mutations reach equilibrium faster than neutral ones), we set the population size to 1000 to simulate the demography of non-African populations (*Balick et al., 2015*; *Speidel et al., 2019*). The advantageous mutation is then

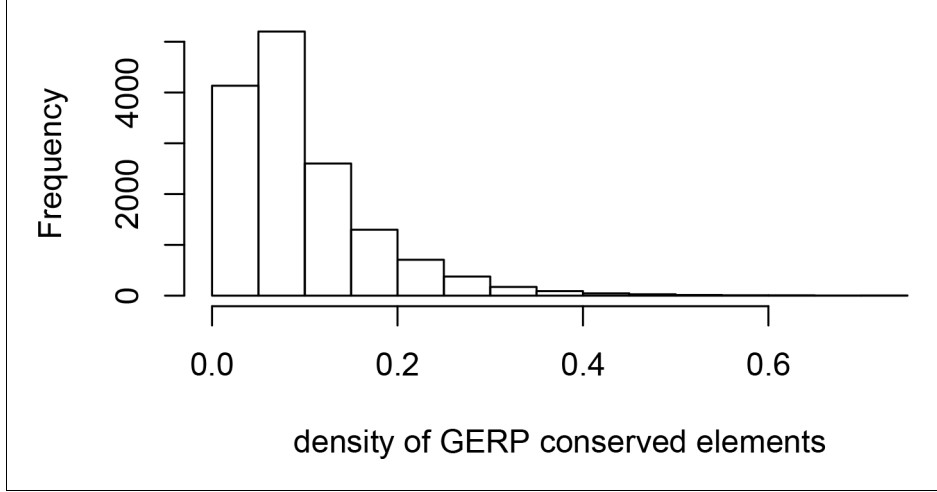

**Figure 9.** Density of GERP conserved elements around genes. The histogram represents the density of GERP conserved elements in 100kb windows centered on Ensembl protein-coding genes.

introduced 500 generations later (~15,000 years later in human evolution) Counting an Out of Africa bottleneck around 60,000 years ago, and if we count that in addition a sweep will take a few more tens of thousands of years to reach frequencies in the iHS or nSL sensitivity range, this brings us within the time window where iHS and nSL still have power (sweeps not older than 30,000 years). We rewind the simulation back to 20,500 generations as many times as necessary until one advantageous mutation goes to fixation. The average number of times we need to rewind the simulations gives us the probability of fixation. We simulate co-dominant advantageous mutations. Each simulation configuration is repeated 1000 times to get the estimates provided in *Table 1*.

The recombination rate is set for the entire one megabase simulated locus at a low value of 0.1 cM/Mb, or ~10% of the average human recombination rate, to maximize the interference effect even during the simulated bottleneck, to see how much a bottleneck can decrease even strong interference.

## Impact of functional differences between disease and non-disease genes on the sweep deficit

The sweep deficit at disease genes could be due to a different representation of gene functions at disease genes compared to control non-disease genes. In this case, disease genes would have less adaptation not because they are disease genes, but because the gene functions that are enriched among disease genes compared to non-disease happen to experience less adaptation. We can test this possibility using Gene Ontology (GO) (*Gene Ontology Consortium and Gene Ontology, 2021*) functional annotations as follows. If GO gene functions that are enriched in disease genes experience less adaptation independently of the disease status of genes, then we can predict that non-disease genes with these functions should also experience less adaptation than non-disease genes that do not have these GO functions. In total, we find that 3,097 GO annotations are enriched in disease genes compared to confounding factors-matched controls (bootstrap test p≤0.01). In our dataset, half of non-disease genes have 20 or more of these GO annotations, and half have less than 20 (very few have none). We find no difference in the sweep prevalence between the two groups (20 or more annotations vs. less than 20 annotations at least 300 kb away; FPR=0.15). The sweep deficit at disease genes is therefore unlikely to be due to the gene functions that are more represented in disease genes compared to controls. In addition, such a scenario would not explain the lack of sweep deficit observed at disease genes with high recombination rates and low numbers of disease variants (*Figure 6*).

## Acknowledgements
We wish to thank Dan Shrider for helpful comments on the results presented in the manuscript.

## Additional information

### Funding

| Funder | Grant reference number | Author |
|---|---|---|
| University of Arizona | startup to David Enard | David Enard |

The funders had no role in study design, data collection and interpretation, or the decision to submit the work for publication.

### Author contributions
Chenlu Di, Conceptualization, Data curation, Formal analysis, Validation, Investigation, Visualization, Writing - original draft, Writing - review and editing; Jesus Murga Moreno, Formal analysis, Methodology; Diego F Salazar-Tortosa, Writing - original draft, Writing - review and editing; M Elise Lauterbur, Writing - review and editing; David Enard, Conceptualization, Data curation, Software, Formal analysis, Supervision, Funding acquisition, Validation, Investigation, Visualization, Methodology, Writing - original draft, Project administration, Writing - review and editing, Interpretation of Results

Author ORCIDs
Diego F Salazar-Tortosa  http://orcid.org/0000-0003-4289-7963
David Enard  https://orcid.org/0000-0003-2634-8016

Decision letter and Author response
Decision letter https://doi.org/10.7554/eLife.69026.sa1
Author response https://doi.org/10.7554/eLife.69026.sa2

## Additional files

### Supplementary files
• Transparent reporting form

### Data availability

The entire article is based on publicly available disease genes and genomic data. The disease genes used and sweep data and the sweep enrichment analysis pipeline (bootstrap test and False Positive risk estimation) with the required input files including the confounding factors are available at https://github.com/DavidPierreEnard/Gene_Set_Enrichment_Pipeline (copy archived at https://archive.softwareheritage.org/swh:1:rev:7b755c0c23dd4d7c3f54c4b53e74366e4041ac8f).

The following previously published datasets were used:

| Author(s) | Year | Dataset title | Dataset URL | Database and Identifier |
|---|---|---|---|---|
| Auton A | 2020 | DisGeNET | https://www.disgenet.org/ | DisGeNET, disgenet.org/ |
| Auton A | 2015 | 1000 Genomes | ftp://ftp.1000genomes.ebi.ac.uk/vol1/ftp/phase3/data | 1000 Genomes Phase 3, 1000genomes.ebi.ac.uk/vol1/ftp/phase3/data |

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
