## [Decision Letter]

**Acceptance summary:**

This paper is of interest for scientists studying human genetic adaptation and disease. Specifically, this study examines whether genes implicated in human Mendelian diseases harbor more or fewer signals of recent positive selection than other genes in the human genome. The work improves on previous studies at addressing this important question by examining a larger datasets of disease genes as well as carefully controlling for confounding factors that could result in disease genes and non-disease genes showing different patterns of genetic variation.

**Decision letter after peer review:**

[Editors’ note: the authors submitted for reconsideration following the decision after peer review. What follows is the decision letter after the first round of review.]

Thank you for resubmitting the paper entitled "Decreased adaptation at human disease genes as a possible consequence of interference between advantageous and deleterious variants" for further consideration by *eLife*. Your revised article has been evaluated by a Senior Editor and a Reviewing Editor. We are sorry to say that we have decided that this submission will not be considered further for publication by *eLife*.

Although both the Reviewers and myself agreed that the questions being posed are interesting, we also agreed that several serious concerns will require extensive additional work. In summary, our key concerns are the following:

1. The fact that the signal is only observed in Africa substantially weakens the main conclusion of the paper. Given how central this is to the entire conclusion of the paper we felt that significant simulation work would be needed to demonstrate that demography likely diminished the effect of interactions in out of Africa populations

2. It is not clear the interference story is the right explanation. The fact that the enrichments are more pronounced in low recombination regions – meant to be evidence in favor of selective interference- could equally be explained by the increased power to detect selective sweeps in low recombining regions.

3. Finally, it is unclear if the selective interference signature is really something specific to Mendelian genes or a general feature of regions of the genome evolving under strong purifying selection. If purifying selection is the driving force, we felt that it would make sense to directly measure that by showing that regions of the genome evolving under strong selective constraints are depleted for selective sweeps (whether or not these regions overlap Mendelian disease genes).

I am sorry that we cannot be more positive at this time but we hope that you will find the reviews helpful and wish you good luck with this work.

*Reviewer #1 (Recommendations for the authors):*

The relationship between genetic disease and adaptation is important for biomedical research as well as understanding human evolution. This topic has received considerable attention over the past several decades in human genetics research. The present manuscript provides a much more comprehensive and rigorous analysis of this topic. Specifically, the authors select a set of ~4000 human Mendelian disease genes and examine patterns of recent positive selection in these genes using the iHS and nSL tests (both haplotype test) for selection. They then compare the signals of sweeps to control genes. Importantly, they match the control set to the disease genes based upon many different genomic variables, such as recombination rate, amount of background selection, expression level, etc. The authors find that there is a deficit of selective sweeps in disease genes. They test several hypotheses for this deficit. They find that the deficit of sweeps is stronger in disease genes at low recombination rate and those that have more disease mutations. From this, the authors conclude that strongly deleterious mutations could be impeding selective sweeps.

Strengths

The manuscript includes a number of important strengths:

1) It tackles an important question in the field. The question of selection in disease genes has been very well-studied in the past, with conflicting viewpoints. The present study examines this topic in a rigorous way and finds a deficit of sweeps in disease genes.

2) The statistical analyses are rigorously done. The genome is a confusing place and there can often be many reasons why a certain set of genes could differ from another set of genes, unrelated to the variable of interest. Di et al., carefully match on these genomic confounders. Thus, they rigorously demonstrate that sweeps are depleted in disease genes relative to control genes. Further, the pipeline for ranking the genes and testing for significance is solid.

3) The Introduction of the manuscript nicely relates different evolutionary models and explanations to patterns that could be seen in the data. As such, the present manuscript isn't just merely an exploratory analysis of patterns of sweeps in disease genes. Rather, it tests specific evolutionary scenarios.

Weaknesses

1) The authors did not discuss or test a basic explanation for the deficit of sweeps in disease genes. Namely, certain types of genes, when mutated, give rise to strong Mendelian phenotypes. However, mutations in these genes do not result in variation that gives rise to a phenotype on which positive selection could occur. In other words, there are just different types of genes underlying disease and positive selection. I could think that such a pattern would be possible if humans are close to the fitness optimum and strong effect mutations (like those in Mendelian disease genes) result in moving further away from the fitness optimum. On the other hand, more weak effect mutations could be either weakly deleterious or beneficial and subject to positive selection. I'm not sure whether these patterns would necessarily be captured by the overall measures of constraint which the disease and non-disease genes were matched on.

2) While I think the authors did a superb job of controlling for genome differences between disease and non-disease genes, the analysis of separating regions by recombination rate and number of disease mutations does not seem as rigorous. Specifically, the authors tested for enrichment of sweeps in disease genes vs control and then stratified that comparison by recombination rate and/or number of disease mutations. While this nicely matches the disease genes to the control genes, it is not clear whether the high recombination rate genes differ in other important attributes from the low recombination rate genes. Thus, I worry whether there could be a confounder that makes it easier/harder to detect an enrichment/deficit of sweeps in regions of low/high recombination.

I have a number of suggestions to improve the presentation:

1) Maybe highlight in the abstract and early on that "disease genes" are "Mendelian disease genes" and do not include genes implicated in complex traits.

2) I had a hard time understanding Figure 2. Maybe consider flipping the order of Figures 2 and 3? I think it's easier to understand Figure 3 as it shows the result more directly. Then, Figure 2 tests the significance of this over different thresholds.

3) Figure 5, it's hard to interpret a "deficit" that goes up. Maybe consider having the bars go lower than 0 to indicate a deficit?

4) It might be good to cite Balick et al., (2015) PLoS Genetics as they examine dominance coefficients in Mendelian disease genes.

*Reviewer #2 (Recommendations for the authors):*

This paper seeks to test the extent to which adaptation via selective sweeps has occurred at disease-associated genes vs genes that have not (yet) been associated with disease. While there is a debate regarding the rate at which selective sweeps have occurred in recent human history, it is clear that some genes have experienced very strong recent selective sweeps. Recent papers from this group have very nicely shown how important virus interacting proteins have been in recent human evolution, and other papers have demonstrated the few instances in which strong selection has occurred in recent human history to adapt to novel environments (e.g. migration to high altitude, skin pigmentation, and a few other hypothesized traits).

One challenge in reading the paper was that I did not realize the analysis was exclusively focused on Mendelian disease genes until much later (the first reference is not until the end of the introduction on pages 7-8 and then not at all again until the discussion, despite referring to "disease" many times in the abstract and throughout the paper). It would be preferred if the authors indicated that this study focused on Mendelian diseases (rather than a broader analysis that included complex or infectious diseases). This is important because there are many different types of diseases and disease genes. Infectious disease genes and complex disease genes may have quite different patterns (as the authors indicate at the end of the introduction).

The abstract states "Understanding the relationship between disease and adaptation at the gene level in the human genome is severely hampered by the fact that we don't even know whether disease genes have experienced more, less, or as much adaptation as non-disease genes during recent human evolution." This seems to diminish a large body of work that has been done in this area. The authors acknowledge some of this literature in the introduction, but it would be worth toning down the abstract, which suggests there has been no work in this area. A review of this topic by Lluis Quintana-Murci1 was cited, but diminished many of the developments that have been made in the intersection of population genetics and human disease biology. Quintana-Murci says "Mendelian disorders are typically severe, compromising survival and reproduction, and are caused by highly penetrant, rare deleterious mutations. Mendelian disease genes should therefore fit the mutation-selection balance model, with an equilibrium between the rate of mutation and the rate of risk allele removal by purifying selection", and argues that positive selection signals should be rare among Mendelian disease genes. Several other examples come to mind. For example, comparing Mendelian disease genes, complex disease genes, and mouse essential genes was the major focus of a 2008 paper2, which pointed out that Mendelian disease genes exhibited much higher rates of purifying selection while complex disease genes exhibited a mixture of purifying and positive selection. This paper was cited, but only in regard to their findings of complex diseases. A similar analysis of McDonald-Kreitman tables was performed around Mendelian disease genes vs non-disease genes, and found "that disease genes have a higher mean probability of negative selection within candidate cis-regulatory regions as compared to non-disease genes, however this trend is only suggestive in EAs, the population where the majority of diseases have likely been characterized". Both of these studies focused on polymorphism and divergence data, which target older instances of selection than iHS and nSL statistics used in the present study (but should have substantial overlap since iHS is not sensitive to very recent selection like the SDS statistic). Regardless, the findings are largely consistent, and I believe warrant a more modest tone.

There are some aspects of the current study that I think are highly valuable. For example, the authors study most of the 1000 Genomes Project populations (though the text should be edited since the admixed and South Asian populations are not analyzed, so all 26 populations are not included, only the populations from Africa, East Asia, and Europe are analyzed; a total of 15 populations are included Figures 2-3). Comparing populations allows the authors to understand how signatures of selection might be shared vs population-specific. Unfortunately, the signals that the authors find regarding the depletion of positive selection at Mendelian disease genes is almost entirely restricted to African populations. The signal is not significant in East Asia or Europe (Figure 2 clearly shows this). It seems that the mean curve of the fold-enrichment as a function of rank threshold (Figure 3) trends downward in East Asian and European populations, but the sampling variance is so large that the bootstrap confidence intervals overlap (1). The paper should therefore revise the sentence "we find a strong depletion in sweep signals at disease genes, especially in Africa" to "only in Africa". This opens the question of why the authors find the particular pattern they find. The authors do point out that a majority of Mendelian disease genes are likely discovered in European populations, so is it that the genes' functions predate the Out-of-Africa split? They most certainly do. It is possible that the larger long-term effective population size of African populations resulted in stronger purifying selection at Mendelian disease genes compared to European and East Asian populations, where smaller effective population sizes due to the Out-of-Africa Bottleneck diminished the signal of most selective sweeps and hence there is little differentiation between categories of genes, "drift noise". It is also surprising to note that the authors find selection signatures at all using iHS in African populations while a previous study using the same statistic could not differentiate signals of selection from neutral demographic simulations4.

The authors find that there is a remarkably (in my view) similar depletion across all but one MeSH disease classes. This suggests that "disease" is likely not the driving factor, but that Mendelian disease genes are a way of identifying where there are strongly selected deleterious variants recurrently arising and preventing positively selected variants. This is a fascinating hypothesis, and is corroborated by the finding that the depletion gets stronger in genes with more Mendelian disease variants. In this sense, the authors are using Mendelian disease genes as a proxy for identifying targets of strong purifying selection, and are therefore not actually studying Mendelian disease genes. The signal could be clearer if the test set is based on the factor that is actually driving the signal.

One of the most important steps that the authors undertake is to control for possible confounding factors. The authors identify 22 possible confounding factors, and find that several confounding factors have different effects in Mendelian disease genes vs non-disease genes. The authors do a great job of implementing a block-bootstrap approach to control for each of these factors. The authors talk specifically about some of these (e.g. PPI), but not others that are just as strong (e.g. gene length). I am left wondering how interactions among other confounding factors could impact the findings of this paper. I was surprised to see a focus on disease variant number, but not a control for CDS length. As I understand it, gene length is defined as the entire genomic distance between the TSS and TES. Presumably genes with larger coding sequence have more potential for disease variants (though number of disease variants discovered is highly biased toward genes with high interest). CDS length would be helpful to correct for things that pS does not correct for, since pS is a rate (controlling for CDS length) and does not account for the coding footprint (hence pS is similar across gene categories).

The authors point out that it is crucial to get the control set right. This group has spent a lot of time thinking about how to define a control set of genes in several previous papers. But it is not clear if complex disease genes and infectious disease genes are specifically excluded or not. Number of virus interactions was included as a confounding factor, so VIPs were presumably not excluded. It is clear that the control set includes genes not yet associated with Mendelian disease, but the focus is primarily on the distance away from known Mendelian disease genes.

1. Quintana-Murci L. Understanding rare and common diseases in the context of human evolution. Genome Biol. 2016 Nov 7;17(1):225. PMCID: PMC5098287

2. Blekhman R, Man O, Herrmann L, Boyko AR, Indap A, Kosiol C, Bustamante CD, Teshima KM, Przeworski M. Natural selection on genes that underlie human disease susceptibility. Curr Biol. Elsevier BV; 2008 Jun 24;18(12):883-889. PMCID: PMC2474766

3. Torgerson DG, Boyko AR, Hernandez RD, Indap A, Hu X, White TJ, Sninsky JJ, Cargill M, Adams MD, Bustamante CD, Clark AG. Evolutionary processes acting on candidate cis-regulatory regions in humans inferred from patterns of polymorphism and divergence. PLoS Genet. Public Library of Science (PLoS); 2009 Aug;5(8):e1000592. PMCID: PMC2714078

4. Granka JM, Henn BM, Gignoux CR, Kidd JM, Bustamante CD, Feldman MW. Limited evidence for classic selective sweeps in African populations. Genetics. Oxford University Press (OUP); 2012 Nov;192(3):1049-1064. PMCID: PMC3522151

Overall, I think the authors exaggerate the novelty of their findings. Also, there is considerable debate regarding the prevalence of selective sweeps in recent human evolution, with many studies (not including the present authors) suggesting selective sweeps have been rare (and instead soft sweeps and other more complicated forms of adaptation likely being much more prevalent)5. The authors of this study previously argued that viral interacting proteins (VIPs) are a major source of the selective sweeps that have occurred in humans. These genes are presumably included in this study, as number of virus interactions is included as a confounder. It is therefore surprising that Mendelian disease genes have such a higher number of virus interactions (indeed, one of the strongest signals in Figure 1), but miss out on the sweep signals.

There have been a number of papers on how confounding factors can drive patterns such as the ones observed in this paper. While the authors do a tremendous job at controlling for confounding factors, I wonder if this depletion is restricted to Mendelian disease genes. Are there other categories that exhibit a similar pattern? It seems the authors should dive deeper into this phenomena, and define the test set of genes based on phyloP or GERP scores (or maybe their "conservation 50kb" confounding factor metric), and see if the depletion in positive selection is more strongly associated with strong purifying selection regions.

It would be helpful to see how similar the confounding factors are for control and Mendelian disease genes are. For example, does each Mendelian diseases gene have a control gene that is similar and others are quite far, or do they tend to be quite different overall?

The authors test for pleiotropy by comparing genes associated with multiple Mendelian diseases vs genes associated with only 1. This should be a quantitative analysis not a dichotomous analysis. There is a lot of biased data here as some genes are more highly studied and may be more highly associated than others. Nonetheless, the conclusion that pleiotropy is likely not a driver of depleted positive selection is probably robust.

5. Schrider DR, Kern AD. Soft sweeps are the dominant mode of adaptation in the human genome. Mol Biol Evol. 2017 Aug 1;34(8):1863-1877. PMCID: PMC5850737

*Reviewer #3 (Recommendations for the authors):*

In this paper, the authors ask whether selective sweeps (as measured by the iHS and nSL statistics) are more or less likely to occur in or near genes associated with Mendelian diseases ("disease genes") than those that are not ("non-disease genes"). The main result put forward by the authors is that genes associated with Mendelian diseases are depleted for sweep signatures, as measured by the iHS and nSL statistics, relative to those which are not.

The evidence for this comes from an empirical randomization scheme to assess whether genes with signatures of a selective sweep are more likely to be Mendelian disease genes that not. The analysis relies on a somewhat complicated sliding threshold scheme that effectively acts to incorporate evidence from both genes with very large iHS/nSL values, as well as those with weaker signals, while upweighting the signal from those genes with the strongest iHS/nSL values. Although I think the anlaysis could be presented more clearly, it does seem like a better analysis than a simple outlier test, if for no other reason than that the sliding threshold scheme can be seen as a way of averaging over uncertainty in where one should set the threshold in an outlier test (along with some further averaging across the two different sweeps statistics, and the size of the window around disease associated genes that the sweep statistics are averaged over). That said, the particular approach to doing so is somewhat arbitrary, but it's not clear that there's a good way to avoid that.

In addition to reporting that extreme values of iHS/nSL are generally less likely at Mendelian disease genes, the authors also report that this depletion is strongest in genes from low recombination regions, or which have >5 specific variants associated with disease.

Drawing on this result, the authors read this evidence to imply that sweeps are generally impeded or slowed in the vicinity of genes associated with Mendelian diseases due to linkage to recessive deleterious variants, which hitchhike to high enough frequencies that the selection against homozygotes becomes an important form of interference. This phenomenon was theoretically characterized by Assaf et al., 2015, who the authors point to for support. That such a phenomenon may be acting systematically to shape the process of adaptation is an interesting suggestion. It's a bit unclear to me why the authors specifically invoke recessive deleterious mutations as an explanation though. Presumably any form of interference could create the patterns they observe? This part of the paper is, as the authors acknowledge, speculative at this point.

I'm also a bit concerned by the fact that the signal is only present in the African samples studied. The authors suggest that this is simply due to stronger drift in the history of European and Asian samples. This could be, but as a reader it's a bit frustrating to have to take this on faith.

There are other analyses that I don't find terribly convincing. For example, one of the anlayses shows that iHS signals are no less depleted at genes associated with >5 diseases than with 1 does little to convince me of anything. It's not particularly clear that # of associated disease for a given gene should predict the degree of pleiotropy experienced by a variant emerging in that gene with some kind of adaptive function. Failure to find any association here might just mean that this is not a particularly good measure of the relevant pleiotropy.

A last parting thought is that it's not clear to me that the authors have excluded the hypothesis that adaptive variants simply arise less often near genes associated with disease. The fact that the signal is strongest in regions of low recombination is meant to be evidence in favor of selective interference as the explanation, but it is also the regime in which sweeps should be easiest to detect, so it may be just that the analysis is best powered to detect a difference in sweep initiation, independent of possible interference dynamics, in that regime.

I'm a bit confused as to why the authors choose to promote the *recessive* deleterious hypothesis specifically. I think the result would be better supported if there were some additional source of information in to indicate that (a) this excess of recessive deleterious variants near disease associated genes *exists*, and (b) that the magnitude of the effect is consistent with the signals being reported.

One part of this paper that was a bit difficult for me to understand at first was the sliding threshold scheme to summarize the strength of the evidence. The procedure is easy enough to understand, but it's not completely obvious at first how to interpret it statistically. After spending some time with it, here is how I understand it.

The sliding threshold scheme amounts to an implicit upweighting of evidence from the genes with the strongest iHS signals, relative to those with lower signals, where the precise degree of upweighting depends on decisions about how many different thresholds to use. To see this, consider the simple case where we look at only the top 20 iHS signals, and we use just two rank thresholds, one at 10, and one at 20. Then, let us write D_{1:10} for the number of disease genes among the top 10 iHS(nSL) signals, and ND_{1:10} for the number of control non-disease genes among the top the top 10 iHS signals, and D_{1:20} and ND_{1:20} for the number of disease and non-disease genes among the top 20. The authors' summary statistic can be written as

D_{1:10} – ND_{1:10} + D_{1:20} – ND_{1:20}

Now, recognizing that D_{1:20} = D_{1:10} + D_{11:20} and ND_{1:20} = ND_{1:10} + ND_{11:20}, where D_{11:20} and ND_{11:20} give the number of disease and non-disease genes respectively among those with iHS(nSL) ranks 11-20, it follows that we can rewrite the authors' summary statistic as

D_{1:10} – ND_{1:10} + (D_{1:10} + D_{11:20}) – (ND_{1:10} + ND_{11:20}) = 2*(D_{1:10} – ND_{1:10}) + 1*(D_{11:20} – ND_{11:20})

which allows us to see the statistic as a sum of the difference in disease vs non-disease genes contained within non-overlapping bins, where each bin is weighted by a term that depends on how far down the list it is. In general, if there are K bins and we assign the first bin corresponding to genes that are below the first rank threshold and index of 1, the second bin, corresponding to genes below the second but above the first, an index of 2, etc., then the summary statistic is given by

sum_k^K (K-K^+^1)*(D_k – ND_k)

where I have slightly abused my notation by writing D_k for the number of disease genes in the ith bin (i.e. D_2 in my general expression corresponds to D_{11:20} in the specific example above).

This formulation makes clearer (at least to me) how the analysis works by upweighting the contributions from more highly ranked iHS(nSL) bins, while still giving some weight to lower bins.

In the text, the authors say that they do this to deliberately avoid an outlier approach, and that this approach "makes less assumptions on the expected strength of selective sweeps" (line 275/276). It's not obvious to me that this is true. Consider: for a particular distribution of "null" iHS(nSL) values at non-sweeps, and a particular distribution for "non-null" iHS(nSL) values at true sweeps, there is presumably *some* choice of weights that would be optimal for detecting the signal the authors are looking for. A standard outlier approach (e.g. look only at the top X iHS(nSL) signals) would amount to assigning equal weights to all of the bins ranked higher than X, and weights of zero to all other bins, while the authors' scheme amounts to a different set of weights that decay as we move from higher bins to lower bins, in a way that depends on exactly how many bins/thresholds are given, and how they are spaced. It is not obvious to me that the authors' scheme makes fewer assumptions, although it clearly makes *different* assumptions.

I have written this out mostly to demistify the analysis pipeline to myself. Having done so, I do feel like I understand it better. I think that from a reader's perspective, the paper would be substantially improved by writing out the methods in clear symbolic notation. The authors don't necessarily have to include my new rewriting of their scheme in terms of non-overlapping bins (though I do think doing so would help clarify the relationship between their analysis and a more standard outlier approach), I think writing it down mathematically or perhaps writing out the algorithm in pseudo-code would help a lot.

It would be helpful if the authors could clarify earlier in the manuscript that their analysis is specifically going to focus on genes associated with Mendelian diseases. Currently, I believe this is not clarified until line 188, and as a reader, it left me feeling pretty unmoored throughout most of the introduction, because I couldn't figure out what "disease genes" really meant.

On line 250/251, the authors write:

"All these factors have been shown to affect adaptation (Methods),"

where "these factors" refers to a list given in the previous sentence. Here, the authors should provide a citation to previous work for this claim.

---

## [Author Response]

[Editors’ note: The authors appealed the original decision. What follows is the authors’ response to the first round of review.]

Although both the Reviewers and myself agreed that the questions being posed are interesting, we also agreed that several serious concerns will require extensive additional work. In summary, our key concerns are the following:1. The fact that the signal is only observed in Africa substantially weakens the main conclusion of the paper. Given how central this is to the entire conclusion of the paper we felt that significant simulation work would be needed to demonstrate that demography likely diminished the effect of interactions in out of Africa populations.

In our response to reviewers, we now explain how some crucial insights provided by the reviewers have now made it possible for us to make a stronger case for the interference between deleterious and advantageous mutations. As is now clearly shown by population simulations, bottlenecks of the same magnitude as the Out of Africa bottleneck, sharply reduce the interference of deleterious variants, and as a result the difference in sweep occurrence between disease and non-disease genes. We believe that this makes the weaker sweep deficit we found in Europe and East Asia not a weakness, but instead a strength, as an expected feature under the explanatory scenario that we propose. The reviewers had the correct intuition here, and we would like to thank them for this, since we never thought of it ourselves before.

Please see our response to the reviewers (especially reviewer 1) for more details on our new results.

2. It is not clear the interference story is the right explanation. The fact that the enrichments are more pronounced in low recombination regions – meant to be evidence in favor of selective interference- could equally be explained by the increased power to detect selective sweeps in low recombining regions.

We would like to apologize for not stating more clearly the implications of a few important results in our manuscript. In point 2, it is proposed that a simple reduction of power to detect sweeps in high recombination regions, or reciprocally an increase of power to detect sweeps in low recombination regions, can explain our results, rather than interference between deleterious and advantageous variants. Here, we believe that we did not insist enough on the fact that a difference in power between low and high recombination, does not explain the different sweep deficits observed with few or many disease variants in low recombination regions. We already stated this important difference, page 16 lines 445 to 457 in the initial manuscript, but should have been more explicit:

“Low recombination is however not sufficient on its own to create a sweep deficit, and we further test if the sweep deficit also depends on the number of disease variants at each disease gene. In our dataset, approximately half of all the disease genes have five or more disease variants, and the other half have four or less disease variants (Methods). In further agreement with possible interference of recessive deleterious variants, the sweep deficit is much more pronounced at disease genes with five or more disease variants (Figure 4, FPR=8.10-4). The sweep deficit at disease genes with four or less disease variants is barely significant compared to control non-disease genes (Figure 4, FPR=0.032). In addition, disease genes with five or more disease variants, but with recombination higher than the median recombination rate, do not have a strong sweep deficit either (Figure 4, FPR=0.026). A higher number of disease variants alone is thus not enough to explain the sweep deficit. In a similar vein, disease genes with a recombination rate less than the median recombination rate, and with four or less disease variants, do not exhibit a strong sweep deficit (Figure 4, FPR=0.021)”.

If there is no interference between deleterious and advantageous variants, then, the sweep deficit at disease genes could be more pronounced in low recombination regions (i) because disease genes experience constitutively less adaptive mutations (alternative explanation notably suggested by reviewer 1), and (ii) because this sweep deficit is more visible in low recombination regions where sweeps are easier to detect in the first place.

Our existing results, together with new results, show that this alternative explanation is unlikely.

First, the higher power to detect sweeps in low recombination regions does not account for the fact that we found a pronounced sweep deficit specifically at disease genes in low recombination regions, AND a high number of disease variants (result from initial manuscript quoted above). The sweep deficit at disease genes in low recombination regions and low numbers of disease variants is only marginally significant. According to the alternative explanation where disease genes have constitutively less adaptive mutations and the difference between low and high recombination regions is just a matter of power, then we should observe a sweep deficit as significant as the sweep deficit at disease genes with a large number of disease variants.

Second, we added new results where we use the fact that nSL is much more robust to different levels of recombination than iHS. As shown in the new figure 6, nSL has much more similar power in low and high recombination regions than iHS. This is not surprising given that nSL was initially designed to be more robust to recombination than iHS. When we use nSL only to measure a sweep deficit at disease genes in low or high recombination regions, we still find the same results as before. A difference in power between low and high recombination regions thus cannot explain our results. We provide these new results P21L590:

“We further find that the difference in sweep deficits between high and low recombination regions is not affected when using only nSL as a sweep statistic (Methods). The nSL statistic was initially designed to be more robust to recombination than iHS (Ferrer-Admetlla et al., 2014), and to have more similar power in low and high recombination regions, and here we confirm this greater robustness. The two distributions of nSL sweep ranks, one for the lower recombination half and one for the higher recombination half of the genes, are much more similar than the two corresponding distributions of iHS sweep ranks (Figure 7A,B). Low recombination regions only have a slight excess of top-ranking nSL signals compared to high recombination regions. Such a small difference is unlikely to generate the substantial discrepancy in power needed to explain the much stronger sweep deficit in low recombination regions. The sweep deficit is substantial when using only nSL on all the disease genes and their controls (Figure 7C; FPR<5.10-4). The nSL-only sweep deficit is only marginally significant in high recombination regions (FPR=0.043, deficit score=- 9,227.4), but strongly significant and about four times more pronounced in low recombination regions (FPR<5.10-4, deficit score=-33,177.2), the same relative difference observed when using both iHS and nSL (Figure 6)”

Third and most importantly, as we also mention in additional detail in our response to point 3, we have conducted McDonald-Kreitman tests (MK test) that show that, on the long term, disease genes do not adapt constitutively less than non-disease genes. A prerequisite for higher power in low recombination regions to explain our results is that the sweep deficit would have to be due to disease genes having constitutively less adaptation, period, whether we look at recent or much more long-term adaptation, which is what the MK test does. As shown in our response to point 3, disease genes do not experience less adaptation than control genes, and even tend to experience more long-term strong adaptation than the genome baseline. This implies that the sweep deficit is therefore more likely to be a transient property due to the presence of segregating, and interfering deleterious variants at this specific, current point in evolutionary time. We now make these points clearer in the manuscript, given that our initial manuscript did not address alternative explanations explicitly and deeply enough (see also our response to point 3):

We now write:

“Disease genes do not experience constitutively less long-term adaptive mutations A deficit of strong recent adaptation (strong enough to affect iHS or 𝑛𝑆!) raises the question of what creates the sweep deficit at disease genes. […] A more transient evolutionary process is thus more likely to explain our results.”

Then: “More importantly, the fact that constitutively less adaptation at disease genes combined to more power to detect sweeps in low recombination regions does not explain our results, is made even clearer by the fact that disease genes in low recombination regions and with many disease variants have in fact experienced more, not less long-term adaptation according to an MK analysis using both ABC-MK and GRAPES (Figure 5F,G,H,I,J). […] This further strengthens the evidence in favor of interference during recent human adaptation: the limiting factor does not seem to be the supply of strongly advantageous variants, but instead the ability of these variants to have generated sweeps recently by rising fast enough in frequency.”

3. Finally, it is unclear if the selective interference signature is really something specific to Mendelian genes or a general feature of regions of the genome evolving under strong purifying selection. If purifying selection is the driving force, we felt that it would make sense to directly measure that by showing that regions of the genome evolving under strong selective constraints are depleted for selective sweeps (whether or not these regions overlap Mendelian disease genes).

To address this criticism, it is first important to list again the controls for purifying selection/selective constraint that we have already implemented and described in the initial manuscript. In addition to all the functional densities that we match between disease and control non-disease genes, we matched the density of conserved elements (from phastCons), the pN/pS ratio, and the background selection B statistic from McVicker et al. The latter B statistic quantifies background selection and therefore reflects the amount of deleterious variants that were removed during recent human evolution. This makes it a measure, and a match of the recent levels of purifying selection/constraint experienced by the disease and control nondisease genes. These controls were mentioned and explained in the Results and in the Methods.

Consequently, if general purifying selection rather than mendelian disease status were to explain our results, it would have to be a residual difference in purifying selection that would remain even after all the matching of measures of constraint that we already did. To address this concern, in our revision we have added a more detailed analysis of the strength of deleterious and advantageous mutations in disease and control non-disease gene coding sequences. This new analysis shows that the controls for purifying selection already implemented and described in the manuscript are sufficient to control not only for the amount of sites under purifying selection, but also for the strength of purifying selection in coding sequences when comparing disease and control non-disease genes. This further supports that our main result is a feature of disease genes rather than a general feature of constrained genes. If there is still any residual difference between the purifying selection experienced by disease genes and their controls, it is likely weak enough to the point of not being sufficient to explain alone the strong sweep depletion that we observed at disease genes compared to matched controls.

“These differences between disease and non-disease genes highlight the need to compare disease genes with control non-disease genes with similar levels of selective constraint. To do this and compare sweeps in mendelian disease genes and non-disease genes that are similar in ways other than being associated with mendelian disease (as described in the Results below, Less sweeps at mendelian disease genes in Africa), we use sets of control nondisease genes that are built by a bootstrap test to match the disease genes in terms of confounding factors (Methods), including the confounding factors that represent measures of selective constraint/purifying selection (density of conserved elements, pN and pS, and McVicker’s B; see Methods). To verify that the measures of selective constraint included indeed control for purifying selection when comparing disease and matched control non-disease genes, we run a maximum likelihood version of the McDonald-Kreitman test called GRAPES (Galtier, 2016), to compare the average selection coefficient of deleterious mutations at disease gene coding sequences, compared to the control non-disease gene coding sequences (Methods). We find that disease genes and their non-disease control genes have undistinguishable average strengths of deleterious variants, suggesting that our controls for selective constraint are sufficient, at least to account for constraint at the coding sequence level (Figure 2; comparison test P=0.37).”.

Purifying selection is therefore already properly controlled for in coding sequences when estimating the sweep deficit in disease genes.

To further verify that purifying selection in the whole region around mendelian disease genes (not only coding) is well controlled for, we measured the sweep deficit again, but this time adding GERP information to the existing confounding factors in the bootstrap matching process. GERP scores directly measure the amount of missing substitutions due to purifying selection. We match both the average GERP score in 50kb and 500kb windows centered on genes, and also the density of GERP conserved elements (GERP scores and conserved elements downloaded from the Sidow lab website). Adding GERP scores makes no difference at all compared to our previous results, and the measured sweep deficits are exactly the same, either using all disease genes, or only disease genes in low recombination and with many disease variants where we measured the strongest sweep deficit. Purifying selection was therefore already properly accounted for.

We now make it clearer:

“Verification of purifying selection controls

To further verify that constraint/purifying selection is properly controlled for when comparing mendelian disease and control non-disease genes, we also add the GERP score, as well as the density of both coding and non-coding conserved elements identified by GERP (Davydov et al., 2010) to the list of matched confounding factors (Methods). The average GERP score in a genomic window estimates the amount of substitutions that never happened during long-term evolution because the said mutations were removed by purifying selection (both in coding and non-coding sequences). The sweep deficit in Africa at disease genes compared to controls is completely unchanged when using GERP or not (Figure 4—figure supplement 1). This shows that the measures of selective constraint already included (Methods) are sufficient to control for selective constraint/purifying selection. For this reason, we do not use GERP further (as explained in the Methods, the larger the number of confounding factors that we match, the lower the power of our approach to detect a sweep enrichment or deficit).”

We also added the corresponding GERP Methods.

Finally, we added a comparison that further shows that purifying selection alone does not explain our results. Instead of comparing sweeps at disease and non-disease genes, we compared sweeps (in Africa) between the 1,000 genes with the highest density of conserved, constrained elements and other genes in the genome (according to the density of phastCons conserved elements in 500kb windows). If purifying selection alone is the factor that drives the sweep deficit at disease genes, then we should see a sweep deficit among the genes with the most conserved, constrained elements compared to other genes in the genome. However, we see no such sweep deficit at genes with a high density of conserved, selectively constrained elements (boostrap test + block randomization of genomes, FPR=0.18). Note that for this comparison we had to remove the matching of confounding factors corresponding to functional and purifying selection densities (new Methods).

These new results are now presented P15L424: “Furthermore, we find that the 1,000 genes in the genome with the highest density of conserved elements do not exhibit any sweep deficit (bootstrap test + block-randomized genomes FPR=0.18; Methods). Association with mendelian diseases, rather than a generally elevated level of selective constraint, is therefore what matters to observe a sweep deficit. What then might explain the sweep deficit at disease genes?”.

The new corresponding Methods are described.

We also point out in the revised manuscript that the fact that long-term protein adaptation (see our response to point 2) is similar between disease genes and controls further excludes that purifying selection alone can explain our results. Indeed, purifying selection is stable over evolutionary time, while the evolutionary process explaining the sweep deficit needs to be transient. We write:

“The fact that the baseline long-term coding adaptation is lower genome-wide, but similarly higher in disease and their control genes, also shows that the matched controls do play their intended role of accounting for confounding factors likely to affect adaptation. The fact that long-term protein adaptation is not lower at disease genes also excludes that purifying selection alone can explain the sweep deficit at disease genes, because purifying selection would then also have decreased long-term adaptation. A more transient evolutionary process is thus more likely to explain our results..”

Together, all these new results confirm that purifying selection is properly controlled for between disease genes and controls, and is therefore not sufficient to explain our results. Here, we believe that we did not explain enough before that just having elevated purifying selection does not explain our results. Purifying selection is needed, but more importantly the presence of deleterious, SEGREGATING disease variants is more important than just purifying selection per se. We should have been more explicit about the implications of these results.

This criticism from point 3 may have also been reinforced by the fact that one of the reviewers believed that we did not control for coding sequence length. However, coding sequence length is already fully controlled for by controlling for coding, CDS density. Finally, always about point 3, we believe that our analysis design centered around disease genes is much more tractable than the proposed broader focus on all constrained genes. Indeed, there is a strong correlation between the amount of purifying selection and the functional content of a gene. This means that when comparing genes with a high amount of constrained sites with genes with a low amount of constrained sites, it would be very difficult to account for the covariation of the functional content that would also happen. Again, our results show that what matters more is the presence of deleterious, segregating disease variants.

These analyses were simple to implement, fast and straightforward. We believe that together with the new results addressing point 1 and with the added clarification and analysis for point 2, that we have sufficient ground for you to reconsider your previous editorial decision.

Below we provide a point-by-point detailed response to all the issues raised by each reviewer, with a reminder to the responses to the three main points above where needed.

Reviewer #1 (Recommendations for the authors):[…]1) The authors did not discuss or test a basic explanation for the deficit of sweeps in disease genes. Namely, certain types of genes, when mutated, give rise to strong Mendelian phenotypes. However, mutations in these genes do not result in variation that gives rise to a phenotype on which positive selection could occur. In other words, there are just different types of genes underlying disease and positive selection. I could think that such a pattern would be possible if humans are close to the fitness optimum and strong effect mutations (like those in Mendelian disease genes) result in moving further away from the fitness optimum. On the other hand, more weak effect mutations could be either weakly deleterious or beneficial and subject to positive selection. I'm not sure whether these patterns would necessarily be captured by the overall measures of constraint which the disease and non-disease genes were matched on.

We thank the reviewer for suggesting that alternative explanation. It is indeed important that we compare it with our own explanation. To rephrase the reviewer’s suggestion, it is possible that disease genes may just have a different distribution of fitness effects of new mutations.

Specifically, mutations in disease genes might have such large effects that they will consistently overshoot the fitness optimum, and thus not get closer to this optimum. This would prevent them from being positively selected. Two predictions can be derived from this potential scenario. First, we can predict a sweep deficit at disease genes, which is what we report. Second, we can also predict that disease genes should exhibit a deficit of older adaptation, not just recent adaptation detected by sweep signals. Indeed, the decrease in adaptation due to (too) large effect mutations would be a generic, intrinsic feature of disease genes regardless of evolutionary time. This means that under this explanation, we expect a test of long-term adaptation such as the McDonald-Kreitman test to also show a deficit at disease genes.

This latter prediction differs from the prediction made by our favored explanation of interference between deleterious and advantageous variants. In this scenario, the sweep deficit at disease genes is caused by the presence of deleterious, and most importantly currently segregating disease variants. Because the presence of the segregating variants is transient during evolution, our explanation does not predict a deficit of long-term adaptation. We can therefore distinguish which explanation (the reviewer’s or ours) is the most likely based on the presence or absence of a long-term adaptation deficit at disease genes.

To test this, we now compare protein adaptation in disease and control genes with two versions of the MK test called ABC-MK and GRAPES (refs). ABC-MK estimates the overall rate of adaptation, and also the rates of weak and strong adaptation,and is based on Approximate Bayesian Computation. GRAPES is based on maximum likelihood. Both ABC-MK and GRPES have shown to provide robust estimates of the rate of protein adaptation thanks to evaluations with forward population simulations (refs). We find no difference in long-term adaptation between disease and control non-disease genes, as shown in new figure 4. This shows that the explanation put forward by the reviewer of an intrinsically different distribution of mutation effects at disease genes is less likely than an interference between currently segregating deleterious variants with recent, but not with older long-term adaptation. We even show in the new figure 4 that disease genes and their controls have more, not less strong long-term adaptation compared to the whole human genome baseline (new figure 4C). Also, disease genes in low recombination regions and with many disease variants have experienced more, not less strong long-term adaptation than their controls. Therefore, far from overshooting the fitness optimum due to stronger fitness effects of mutations, it looks like that these stronger fitness effects might in fact be more frequently positively selected in these disease genes.

We now provide these new results:

“Disease genes do not experience constitutively less long-term adaptive mutations A deficit of strong recent adaptation (strong enough to affect iHS or 𝑛𝑆!) raises the question of what creates the sweep deficit at disease genes. […] A more transient evolutionary process is thus more likely to explain our results.”

Then:

“More importantly, the fact that constitutively less adaptation at disease genes combined to more power to detect sweeps in low recombination regions does not explain our results, is made even clearer by the fact that disease genes in low recombination regions and with many disease variants have in fact experienced more, not less long-term adaptation according to an MK analysis using both ABC-MK and GRAPES (Figure 5F,G,H,I,J). ABC-MK in particular finds that there is a significant excess of long-term strong adaptation (Figure 4H, P<0.01) in disease genes with low recombination and with many disease variants, compared to controls, but similar amounts of weak adaptation (Figure 5G, P=0.16). It might be that disease genes with many disease variants are genes with more mutations with stronger effects that can generate stronger positive selection. The potentially higher supply of strongly advantageous variants at these disease genes makes it all the more notable that they have a very strong sweep deficit in recent evolutionary times. This further strengthens the evidence in favor of interference during recent human adaptation: the limiting factor does not seem to be the supply of strongly advantageous variants, but instead the ability of these variants to have generated sweeps recently by rising fast enough in frequency.”

2) While I think the authors did a superb job of controlling for genome differences between disease and non-disease genes, the analysis of separating regions by recombination rate and number of disease mutations does not seem as rigorous. Specifically, the authors tested for enrichment of sweeps in disease genes vs control and then stratified that comparison by recombination rate and/or number of disease mutations. While this nicely matches the disease genes to the control genes, it is not clear whether the high recombination rate genes differ in other important attributes from the low recombination rate genes. Thus, I worry whether there could be a confounder that makes it easier/harder to detect an enrichment/deficit of sweeps in regions of low/high recombination.

We thank the reviewer for emphasizing the need for more controls when comparing our results in low or high recombination regions. We have now compared the confounding factors between low recombination disease genes and high recombination disease genes, as classified in the manuscript. As shown in new supp table Figure 6 figure supplement 1, confounding factors do not differ substantially between low and high recombination disease genes, and are all within a range of +/- 25% of each other. It would take a larger difference for any confounding factor to explain the sharp sweep deficit difference observed between the low and high recombination disease genes. The only factor with a 35% difference between low and high recombination mendelian disease genes is McVicker’s B, but this is completely expected; B is expected to be lower in low recombination regions.

We now write P20L569: “Further note that only moderate differences in confounding factors between low and high recombination mendelian disease genes are unlikely to explain the sweep deficit difference (Figure 6—figure supplement 1).”

Regarding the potential confounding effect of statistical power to detect sweeps differing in low and high recombination regions, please see our earlier response to main point 2.

I have a number of suggestions to improve the presentation:1) Maybe highlight in the abstract and early on that "disease genes" are "Mendelian disease genes" and do not include genes implicated in complex traits.

We apologize for the confusion. It was a mistake to not make it clearer from the start of the manuscript that we focused on mendelian, non-infectious disease genes. We have now modified the title, as well as multiple early mentions of disease genes to make sure that it is immediately clear that we focus on mendelian disease genes.

2) I had a hard time understanding Figure 2. Maybe consider flipping the order of Figures 2 and 3? I think it's easier to understand Figure 3 as it shows the result more directly. Then, Figure 2 tests the significance of this over different thresholds.

Again we are very sorry about the confusion. Following reviewer 3’s recommendations, we have now added a more complete formal description of the statistic we use to measure the false positive risk (FPR). This should make it easier to understand how me go from previous figure 3 to previous figure 2. Based on the reviewer’s recommendation we have now swapped the order of the two figures.

3) Figure 5, it's hard to interpret a "deficit" that goes up. Maybe consider having the bars go lower than 0 to indicate a deficit?

We have modified the figure accordingly. It is now figure 7.

4) It might be good to cite Balick et al., (2015) PLoS Genetics as they examine dominance coefficients in Mendelian disease genes.

We cannot thank the reviewer enough for mentioning that we should cite Balick et al. (Plos Genetics 2015). We were not aware of this paper, and the results presented in it have been invaluable to better explain and strengthen our results. Indeed, the results from Balick et al., likely explain why we see a strong sweep deficit in Africa but not in Europe or East Asia. Balick et al., show that after a bottleneck of the same order of magnitude as observed in Europe and East Asia after the Out of Africa migration (~10 fold reduction in effective population size), there is a sharp decline of the number and impact of segregating recessive deleterious variants. First, many low frequency, segregating recessive deleterious variants disappear when the bottleneck first happens. Second, the remaining recessive deleterious variants are not as deleterious due to the reduced population size and increased genetic drift. Third, it takes a long time for new recessive deleterious variants to reach frequencies high enough that they can occur as homozygotes and be selected against as a result.

These results were obtained by Balick et al., using analytical results and results from simulations where the ancestral African population size was set at 10,000, and the bottlenecked Out of Africa population was set at 1,000. These numbers happen to be very similar to the most recent estimates of these ancestral population sizes by Relate (Speidel et al., 2019). Under such a bottleneck, Balick et al., showed that there should be a substantial decrease of the burden of segregating recessive deleterious mutations (especially the rare ones that are removed by a bottleneck). They also show that additive or dominant deleterious variants are much less affected than recessive deleterious ones. In any case, dominant variants are not expected to induce interference with advantageous variants (Assaf et al., 2015).

We have now added the results of population simulations of loci with recessive deleterious mutations that show that during a bottleneck from a 10,000 to a 1,000 population size, the impact of interference of recessive deleterious variants on advantageous variants is very strongly reduced, even at loci with many recessive deleterious variants initially. First, this result is consistent with the results of Balick et al., (2015). If the burden of recessive deleterious variants is reduced by the bottleneck, then we expect their interference effect to also be reduced. Second, this result is very consistent with our observations that the most bottlenecked population we looked at, East Asia, also happens to be the one with the least significant sweep deficit. Based on these new simulations results, our explanation of interference therefore predicts the observed patterns across different human populations. Note that it predicts the difference between African and non-African populations without requiring our previous, less robust explanation of more false sweep signals in East Asia and Europe.

As shown by our simulation results, the interference effect is reduced strongly enough during an Out of Africa-like bottleneck that it is likely not necessary at all to invoke a higher rate of false positive sweep signals out of Africa to further explain the weaker sweep deficit in Europe and East Asia.

We are now making all these points clear in the revised manuscript. We want to thank the reviewer again for their remarkable intuition that the Balick et al., reference would be useful. We feel that it provided an important missing piece to our manuscript.

We now write:

“Decreased interference of recessive deleterious mutations during a bottleneck may explain the weaker sweep deficit in East Asia and Europe An important observation in our analysis, that any potential explanation needs to account for, is the much weaker sweep deficit at disease genes in Europe and especially in East Asia, compared to Africa. […] This further supports the idea that interference with recessive deleterious variants may explain our observation of a strong sweep deficit at disease genes in Africa, and of weaker sweep deficits out of Africa.”

We also add the corresponding Methods.

Reviewer #2 (Recommendations for the authors):This paper seeks to test the extent to which adaptation via selective sweeps has occurred at disease-associated genes vs genes that have not (yet) been associated with disease. While there is a debate regarding the rate at which selective sweeps have occurred in recent human history, it is clear that some genes have experienced very strong recent selective sweeps. Recent papers from this group have very nicely shown how important virus interacting proteins have been in recent human evolution, and other papers have demonstrated the few instances in which strong selection has occurred in recent human history to adapt to novel environments (e.g. migration to high altitude, skin pigmentation, and a few other hypothesized traits).One challenge in reading the paper was that I did not realize the analysis was exclusively focused on Mendelian disease genes until much later (the first reference is not until the end of the introduction on pages 7-8 and then not at all again until the discussion, despite referring to "disease" many times in the abstract and throughout the paper). It would be preferred if the authors indicated that this study focused on Mendelian diseases (rather than a broader analysis that included complex or infectious diseases). This is important because there are many different types of diseases and disease genes. Infectious disease genes and complex disease genes may have quite different patterns (as the authors indicate at the end of the introduction).

We want to apologize profusely for this avoidable mistake. We have now made it clearer from the very start of the manuscript that we focus on mendelian non-infectious disease genes. We have modified the title and the abstract accordingly, specifying mendelian and non-infectious as required.

The abstract states "Understanding the relationship between disease and adaptation at the gene level in the human genome is severely hampered by the fact that we don't even know whether disease genes have experienced more, less, or as much adaptation as non-disease genes during recent human evolution." This seems to diminish a large body of work that has been done in this area. The authors acknowledge some of this literature in the introduction, but it would be worth toning down the abstract, which suggests there has been no work in this area. A review of this topic by Lluis Quintana-Murci1 was cited, but diminished many of the developments that have been made in the intersection of population genetics and human disease biology. Quintana-Murci says "Mendelian disorders are typically severe, compromising survival and reproduction, and are caused by highly penetrant, rare deleterious mutations. Mendelian disease genes should therefore fit the mutation-selection balance model, with an equilibrium between the rate of mutation and the rate of risk allele removal by purifying selection", and argues that positive selection signals should be rare among Mendelian disease genes. Several other examples come to mind. For example, comparing Mendelian disease genes, complex disease genes, and mouse essential genes was the major focus of a 2008 paper2, which pointed out that Mendelian disease genes exhibited much higher rates of purifying selection while complex disease genes exhibited a mixture of purifying and positive selection. This paper was cited, but only in regard to their findings of complex diseases. A similar analysis of McDonald-Kreitman tables was performed around Mendelian disease genes vs non-disease genes, and found "that disease genes have a higher mean probability of negative selection within candidate cis-regulatory regions as compared to non-disease genes, however this trend is only suggestive in EAs, the population where the majority of diseases have likely been characterized". Both of these studies focused on polymorphism and divergence data, which target older instances of selection than iHS and nSL statistics used in the present study (but should have substantial overlap since iHS is not sensitive to very recent selection like the SDS statistic). Regardless, the findings are largely consistent, and I believe warrant a more modest tone.

We thank the reviewer for their recommendation. We should have written more about what is currently well known or unknown about recent adaptation in disease genes, and in more nuanced terms. Instead of writing “Understanding the relationship between disease and adaptation at the gene level in the human genome is severely hampered by the fact that we don't even know whether disease genes have experienced more, less, or as much adaptation as non-disease genes during recent human evolution”, we now write in the new abstract:

“Despite our expanding knowledge of gene-disease associations, and despite the medical importance of disease genes, their recent evolution has not been thoroughly studied across diverse human populations. In particular, recent genomic adaptation at disease genes has not been characterized as well as long-term purifying selection and long-term adaptation. Understanding the relationship between disease and adaptation at the gene level in the human genome is hampered by the fact that we don’t know whether disease genes have experienced more, less, or as much adaptation as non-disease genes during the last ~50,000 years of recent human evolution.”

We also toned down the start of the introduction. We now write:

“Despite our expanding knowledge of mendelian disease gene associations, and despite the fact that multiple evolutionary processes might connect disease and genomic adaptation at the gene level, these connections are yet to be studied more thoroughly, especially in the case of recent genomic adaptation.”

Although we agree that others have made extensive efforts to characterize older adaptation or purifying selection at disease genes compared to non-disease genes, we still believe that our results are novel and more conclusive about recent positive selection. Our initial statement was however poorly phrased. To our knowledge, our study is the first to look at the issue using specifically sweep statistics that have been shown to be robust to background selection, while also controlling for confounding factors. These sweep statistics have sensitivity for selection events that occurred in the past 30,000 or at most 50,000 years of human evolution (Sabeti et al., 2006). This is a very different time scale compared to the millions of years of adaptation (since divergence between humans and chimpanzees) captured by MK approaches.

We also want to note that we did cite the Blekhman et al., paper for their result of stronger purifying selection in our initial manuscript. It is true however that we did not specify mendelian disease genes, which was confusing. We want to apologize again for it:

From the earlier manuscript: “Multiple recent studies comparing evolutionary patterns between human disease and non-disease genes have found that disease genes are more constrained and evolve more slowly (lower ratio of nonsynonymous to synonymous substitution rate, dN/dS, in disease genes) (Blekhman et al., 2008; Park et al., 2012; Spataro et al., 2017)”

“Among other confounding factors, it is particularly important to take into account evolutionary constraint, i.e the level of purifying selection experienced by different genes. A common intuition is that disease genes may exhibit less adaptation because they are more constrained (Blekhman et al., 2008)”

It is important to remember that, as we mention in the introduction, previous comparisons did not take potential confounding factors at all into account. It is therefore unclear whether their conclusions were specific to disease genes, or due to confounding factors. We have now made this point clearer in the introduction, as we believe that we have made a substantial effort to control for confounding factors, and that it is a substantial departure from previous efforts:

“In contrast with previous studies, we systematically control for a large number of confounding factors when comparing recent adaptation in human mendelian disease and nondisease genes, including evolutionary constraint, mutation rate, recombination rate, the proportion of immune or virus-interacting genes, etc. (please refer to Methods for a full list of the confounding factors included).”.

“These differences between disease and non-disease genes highlight the need to compare disease genes with control non-disease genes with similar levels of selective constraint. To do this and compare sweeps in mendelian disease genes and non-disease genes that are similar in ways other than being associated with mendelian disease (as described in the Results below, Less sweeps at mendelian disease genes), we use sets of control non-disease genes that are built by a bootstrap test to match the disease genes in terms of confounding factors (Methods)”.

Furthermore, we have now added a comparison of older adaptation in disease and non-disease genes using a recent version of the MK test called ABC-MK, that can take background selection and other biases such as segregating weakly advantageous variants into account. Also controlling for confounding factors, we find no difference in older adaptation between disease and non-disease genes (please see our response to main point 2).

Therefore, contrary to the reviewer’s claim that the sweep statistics and MK approaches should have substantial overlap, we now show that it is clearly not the case. We further show that the lack of overlap is expected under our explanation of our results based on interference between recessive deleterious and advantageous variants (see our responses to main point 1 and to reviewer 1 weakness 1).

Previous analyses were using much smaller mendelian disease gene datasets, less recent polymorphism datasets and, critically, did not control for confounding factors. We also note that reference (Torgerson et al., Plos Genetics 2009) does not make any claim about recent positive selection in mendelian disease genes compared to other genes. Their dataset at the time also only included 666 mendelian disease genes, versus the ~4,000 currently known. In short, we do think that we have a claim for novelty, but the reviewer is entirely right that we did a poor job of giving due credit to previous important work. These previous studies deserved much better credit than no credit at all. We want to thank the reviewer from avoiding us the embarrassment of not citing important work.

We now cite the papers referenced by the reviewer as appropriate in the introduction, based on the scope of their results:

“Multiple recent studies comparing evolutionary patterns between human mendelian disease and non-disease genes have found that mendelian disease genes are more constrained and evolve more slowly (Blekhman et al., 2008; Quintana-Murci, 2016; Spataro et al., 2017; Torgerson et al., 2009). An older comparison by Smith and Eyre-Walker (Smith and Eyre-Walker, 2003) found that disease genes evolve faster than non-disease genes, but we note that the sample of disease genes used at the time was very limited.”

“Among possible confounding factors, it is particularly important to take into account evolutionary constraint, i.e the level of purifying selection experienced by different genes. A common intuition is that mendelian disease genes may exhibit less adaptation because they are more constrained (Blekhman et al., 2008; Spataro et al., 2017; Torgerson et al., 2009),”

There are some aspects of the current study that I think are highly valuable. For example, the authors study most of the 1000 Genomes Project populations (though the text should be edited since the admixed and South Asian populations are not analyzed, so all 26 populations are not included, only the populations from Africa, East Asia, and Europe are analyzed; a total of 15 populations are included Figures 2-3). Comparing populations allows the authors to understand how signatures of selection might be shared vs population-specific. Unfortunately, the signals that the authors find regarding the depletion of positive selection at Mendelian disease genes is almost entirely restricted to African populations. The signal is not significant in East Asia or Europe (Figure 2 clearly shows this). It seems that the mean curve of the fold-enrichment as a function of rank threshold (Figure 3) trends downward in East Asian and European populations, but the sampling variance is so large that the bootstrap confidence intervals overlap (1). The paper should therefore revise the sentence "we find a strong depletion in sweep signals at disease genes, especially in Africa" to "only in Africa". This opens the question of why the authors find the particular pattern they find. The authors do point out that a majority of Mendelian disease genes are likely discovered in European populations, so is it that the genes' functions predate the Out-of-Africa split? They most certainly do. It is possible that the larger long-term effective population size of African populations resulted in stronger purifying selection at Mendelian disease genes compared to European and East Asian populations, where smaller effective population sizes due to the Out-of-Africa Bottleneck diminished the signal of most selective sweeps and hence there is little differentiation between categories of genes, "drift noise". It is also surprising to note that the authors find selection signatures at all using iHS in African populations while a previous study using the same statistic could not differentiate signals of selection from neutral demographic simulations.

We want to thank the reviewer profusely for putting us on the right track thanks to their insightful suggestion. As described in our response to reviewer 1 weakness 1, we have now shown with simulations that the interference of deleterious variants on advantageous variants is strongly decreased during a bottleneck of a magnitude similar to the Out of Africa bottlenecks experienced by East Asian and European populations. This decrease of interference is likely strong enough to not require any other explanation, even if other processes may also be at work, such as a decrease of the sweeps signals as suggested by the reviewer.

About the Granka et al., paper, the last author of the current manuscript has already shown in a previous paper (ref) that the type of approaches used to quantify recent adaptation is likely to be severely underpowered due to a number of confounding factors, notably including comparing genic and non-genic windows that are not sufficiently far from each other to not overlap the same sweep signals. Our result are also based on much more recent and less biased sets of SNPs used to measure the sweeps statistics.

The authors find that there is a remarkably (in my view) similar depletion across all but one MeSH disease classes. This suggests that "disease" is likely not the driving factor, but that Mendelian disease genes are a way of identifying where there are strongly selected deleterious variants recurrently arising and preventing positively selected variants. This is a fascinating hypothesis, and is corroborated by the finding that the depletion gets stronger in genes with more Mendelian disease variants. In this sense, the authors are using Mendelian disease genes as a proxy for identifying targets of strong purifying selection, and are therefore not actually studying Mendelian disease genes. The signal could be clearer if the test set is based on the factor that is actually driving the signal.

Based on the reviewer’s comment, we have now better explained why our results are unlikely to be a generic property of purifying selection alone. As we explain in our response to main point 3, our results cannot be explained by purifying selection alone, because we match purifying selection between disease genes and the controls. Indeed, we now show with additional MK analyses and GERP-based analyses that our controls for confounding factors already account for purifying selection. This is shown by the fact that disease genes and their controls have similar distributions of deleterious fitness effects.

In addition, we added a comparison that shows that purifying selection alone does not explain our results. Instead of comparing sweeps at disease and non-disease genes, we compared sweeps (in Africa) between the 1,000 genes with the highest density of conserved, constrained elements and other genes in the genome. If purifying selection is the factor that drives the sweep deficit at disease genes, then we should see a sweep deficit among the genes with the most conserved, constrained elements compared to other genes in the genome. However, we see no such sweep deficit at genes with a high density of conserved, selectively constrained elements (boostrap test + block randomization of genomes, FPR=0.18). See P15L424. Note that for this comparison we had to remove the matching of confounding factors corresponding to functional and purifying selection densities (new Methods P40L1131).

Again, our results are better explained not just by purifying selection alone, but more specifically by the presence of interfering, segregating deleterious variants. It is perfectly possible to have highly constrained parts of the genome without having many deleterious segregating variants at a given time in evolution.

The similarity across MeSH classes can be readily explained if what matters is interference with deleterious segregating variants. Because all types of diseases have deleterious segregating variants, then it is not surprising that different MeSH disease categories have a similar sweep deficit. We make that point clearer in the revised manuscript:

“The sweep deficit is comparable across MeSH disease classes (Figure 8), suggesting that the evolutionary process at the origin of the sweep deficit is not disease specific. This is compatible with a non-disease specific explanation such as recessive deleterious variants interfering with adaptive variants, irrespective of the specific disease type.”.

One of the most important steps that the authors undertake is to control for possible confounding factors. The authors identify 22 possible confounding factors, and find that several confounding factors have different effects in Mendelian disease genes vs non-disease genes. The authors do a great job of implementing a block-bootstrap approach to control for each of these factors. The authors talk specifically about some of these (e.g. PPI), but not others that are just as strong (e.g. gene length). I am left wondering how interactions among other confounding factors could impact the findings of this paper. I was surprised to see a focus on disease variant number, but not a control for CDS length. As I understand it, gene length is defined as the entire genomic distance between the TSS and TES. Presumably genes with larger coding sequence have more potential for disease variants (though number of disease variants discovered is highly biased toward genes with high interest). CDS length would be helpful to correct for things that pS does not correct for, since pS is a rate (controlling for CDS length) and does not account for the coding footprint (hence pS is similar across gene categories).

Based on our response to the previous point, it is clear that a high density of coding sequences, or conserved constrained sequence in general are not enough to explain our results.

Furthermore, we want to remind the reviewer that we already control for coding sequence length through controlling for coding density, since we use windows of constant sizes.

The authors point out that it is crucial to get the control set right. This group has spent a lot of time thinking about how to define a control set of genes in several previous papers. But it is not clear if complex disease genes and infectious disease genes are specifically excluded or not. Number of virus interactions was included as a confounding factor, so VIPs were presumably not excluded. It is clear that the control set includes genes not yet associated with Mendelian disease, but the focus is primarily on the distance away from known Mendelian disease genes.

We are sorry that we were not more explicit from the start of the manuscript. We now make it clearer what the set disease genes includes or not throughout the entire manuscript, by repeating that we focus specifically on mendelian, non-infectious disease genes. By noninfectious, we mean that we excluded genes with known infectious disease-associated variants. This does not exclude most virus-interacting genes since most of them are not associated at the genetic variant level with infectious diseases. It is also important to note that the effect of virus interactions is accounted for by matching the number of interacting viruses between mendelian disease genes and controls. We write: “By non-infectious, we mean that we excluded genes with known infectious disease-associated variants. This does not exclude most VIPs since most of them are not associated at the genetic variant level with infectious diseases. It is important to note that the effect of virus interactions is accounted for by matching the number of interacting viruses between mendelian disease genes and controls.”

Overall, I think the authors exaggerate the novelty of their findings. Also, there is considerable debate regarding the prevalence of selective sweeps in recent human evolution, with many studies (not including the present authors) suggesting selective sweeps have been rare (and instead soft sweeps and other more complicated forms of adaptation likely being much more prevalent)5. The authors of this study previously argued that viral interacting proteins (VIPs) are a major source of the selective sweeps that have occurred in humans. These genes are presumably included in this study, as number of virus interactions is included as a confounder. It is therefore surprising that Mendelian disease genes have such a higher number of virus interactions (indeed, one of the strongest signals in Figure 1), but miss out on the sweep signals.

We have shown in previous work that studies concluding that sweeps were rare in the human genome were missing important controls in their comparisons, that were masking the investigated, expected genome-wide signatures of selective sweeps. Recently, other researchers have also used powerful machine learning approaches that show that selective sweeps are not rare in the human genome. Previous approaches were underpowered, and again were not controlling for important confounding factors.

In addition, because we match the number of interactions with viruses between disease genes and their controls, the higher prevalence of VIPs among disease genes is already controlled for in our results.

There have been a number of papers on how confounding factors can drive patterns such as the ones observed in this paper. While the authors do a tremendous job at controlling for confounding factors, I wonder if this depletion is restricted to Mendelian disease genes. Are there other categories that exhibit a similar pattern? It seems the authors should dive deeper into this phenomena, and define the test set of genes based on phyloP or GERP scores (or maybe their "conservation 50kb" confounding factor metric), and see if the depletion in positive selection is more strongly associated with strong purifying selection regions.

We thank the reviewer for this recommendation, as this analysis is important but was missing in our manuscript. As we already mentioned above, our comparison of genes with a high density of conserved elements with the rest of the genome shows that purifying selection is not enough to create a sweep deficit.

It would be helpful to see how similar the confounding factors are for control and Mendelian disease genes are. For example, does each Mendelian diseases gene have a control gene that is similar and others are quite far, or do they tend to be quite different overall?

We are sorry that the Methods were not clear enough on how we match confounding factors between disease genes and their controls. We now make it more explicit in the Methods what previous paper describes the matching algorithm in detail, and we also provide a brief additional explanation P34L953:

“We choose non-disease control genes that have the same confounding factors characteristics as disease genes (for example, control non-disease genes that have the same gene expression level across tissues as disease genes). The precise matching algorithm is detailed in Enard and Petrov, 2020. In brief, the bootstrap test builds sets of control genes that have the same overall average values for confounding factors as disease genes. For example, the bootstrap test can build 100 control sets, with each set having the same overall average GC content as disease genes. Note that this means that disease genes are not individually matched one by one with one control gene that happens to have the same GC content. Matching genes individually, instead of matching the overall gene sets averages, would indeed limit the pool of potential control genes too drastically. For more details on this, please refer to (Enard and Petrov, 2020).”

The matching procedure is explained in the cited previous paper, but it is indeed important to remind the broad details of how it works. We apologize for the confusion.

The authors test for pleiotropy by comparing genes associated with multiple Mendelian diseases vs genes associated with only 1. This should be a quantitative analysis not a dichotomous analysis. There is a lot of biased data here as some genes are more highly studied and may be more highly associated than others. Nonetheless, the conclusion that pleiotropy is likely not a driver of depleted positive selection is probably robust.

We agree with the reviewer that ideally we would investigate pleiotropy over a continuum in a quantitative regression analysis instead of a dichotomous comparison. The new results showing no deficit of long-term protein adaptation in mendelian disease genes imply that pleiotropy or any other factor decreasing adaptation constitutively (and thus long-term) are unlikely to explain our results. Please refer to our detailed response to main point 2. Consequently, we have removed the part of the initial manuscript that was focusing on pleiotropy through the number of diseases. It was indeed a weaker part in our manuscript.

Reviewer #3 (Recommendations for the authors):In this paper, the authors ask whether selective sweeps (as measured by the iHS and nSL statistics) are more or less likely to occur in or near genes associated with Mendelian diseases ("disease genes") than those that are not ("non-disease genes"). The main result put forward by the authors is that genes associated with Mendelian diseases are depleted for sweep signatures, as measured by the iHS and nSL statistics, relative to those which are not.The evidence for this comes from an empirical randomization scheme to assess whether genes with signatures of a selective sweep are more likely to be Mendelian disease genes that not. The analysis relies on a somewhat complicated sliding threshold scheme that effectively acts to incorporate evidence from both genes with very large iHS/nSL values, as well as those with weaker signals, while upweighting the signal from those genes with the strongest iHS/nSL values. Although I think the anlaysis could be presented more clearly, it does seem like a better analysis than a simple outlier test, if for no other reason than that the sliding threshold scheme can be seen as a way of averaging over uncertainty in where one should set the threshold in an outlier test (along with some further averaging across the two different sweeps statistics, and the size of the window around disease associated genes that the sweep statistics are averaged over). That said, the particular approach to doing so is somewhat arbitrary, but it's not clear that there's a good way to avoid that.In addition to reporting that extreme values of iHS/nSL are generally less likely at Mendelian disease genes, the authors also report that this depletion is strongest in genes from low recombination regions, or which have >5 specific variants associated with disease.Drawing on this result, the authors read this evidence to imply that sweeps are generally impeded or slowed in the vicinity of genes associated with Mendelian diseases due to linkage to recessive deleterious variants, which hitchhike to high enough frequencies that the selection against homozygotes becomes an important form of interference. This phenomenon was theoretically characterized by Assaf et al., 2015, who the authors point to for support. That such a phenomenon may be acting systematically to shape the process of adaptation is an interesting suggestion. It's a bit unclear to me why the authors specifically invoke recessive deleterious mutations as an explanation though. Presumably any form of interference could create the patterns they observe? This part of the paper is, as the authors acknowledge, speculative at this point.

We thank the reviewer for their comments. We are sorry that we did not provide a clear explanation of why only recessive deleterious mutations are expected to interfere more than other types of deleterious variants. This was shown by Assaf et al., (2015), and we should have stated it explicitly. The reason why recessive deleterious variants interfere more than additive or dominant ones is that they can hitchhike together with an adaptive variant to substantial frequencies before negative selection actually happens, when a significant number of homozygous individuals for the deleterious mutation start happening in the population. On the contrary dominant mutations do not make it to the same high frequencies linked to an adaptive variant, because they start being selected negatively as soon as they appear in the population.

We now write: “In diploid species including humans, recessive deleterious mutations specifically have been shown to have the ability to slow down, or even stop the frequency increase of advantageous mutations that they are linked with (Assaf et al., 2015). Dominant variants do not have the same interfering ability, because they do not increase in frequency in linkage with advantageous variants as much as recessive deleterious do, before the latter can be “seen” by purifying selection when enough homozygous individuals emerge in a population (Assaf et al., 2015).”

We have also confirmed with SLiM forward simulations that recessive deleterious variants interfere with adaptive variants much more than dominant ones (Table 1).

I'm also a bit concerned by the fact that the signal is only present in the African samples studied. The authors suggest that this is simply due to stronger drift in the history of European and Asian samples. This could be, but as a reader it's a bit frustrating to have to take this on faith.

We thank the reviewer for pointing out this issue with our manuscript. We have now shown, as detailed above in our response to main point 1, reviewer 1 weakness 1, that a weaker sweep deficit at disease genes in Europe and East Asia is an expected feature under the interference explanation, due to the weakened interference of recessive deleterious variants during bottlenecks of the magnitude observed in Europe and East Asia. We therefore believe that these new results strengthen our previous claim regarding the role interference between deleterious and advantageous variants. We want to thank the reviewer for forcing us to examine the difference between results in Africa and out of Africa, as the manuscript is now more consistent and our results substantially better explained.

There are other analyses that I don't find terribly convincing. For example, one of the anlayses shows that iHS signals are no less depleted at genes associated with >5 diseases than with 1 does little to convince me of anything. It's not particularly clear that # of associated disease for a given gene should predict the degree of pleiotropy experienced by a variant emerging in that gene with some kind of adaptive function. Failure to find any association here might just mean that this is not a particularly good measure of the relevant pleiotropy.

We agree with the reviewer that the number of associated disease may not be a good measure of pleiotropy. Unfortunately to our knowledge there is currently no good measure of gene pleiotropy in human genomes. Given that the evidence in favor of interference of deleterious variants is now strengthened, we have chosen to remove this analysis from the manuscript. As we now explain throughout the manuscript, pleiotropy is an unlikely explanation in the first place because of the fact that disease genes have not experienced less long-term adaptation (see the details on our new MK test results in the response to main point 2).

“We find that overall, disease and control non-disease genes have experienced similar rates of protein adaptation during millions of years of human evolution, as shown by very similar estimated proportions of amino acid changes that were adaptive (Figure 5A,B,C,D,E). This result suggests that disease genes do not have constitutively less adaptive mutations. This implies that processes stable over evolutionary time such as pleiotropy, or a tendency to overshoot the fitness optimum, are unlikely to explain the sweep deficit at disease genes.”.

A last parting thought is that it's not clear to me that the authors have excluded the hypothesis that adaptive variants simply arise less often near genes associated with disease. The fact that the signal is strongest in regions of low recombination is meant to be evidence in favor of selective interference as the explanation, but it is also the regime in which sweeps should be easiest to detect, so it may be just that the analysis is best powered to detect a difference in sweep initiation, independent of possible interference dynamics, in that regime.

We thank the reviewer for stating these important alternative explanations that needed more attention in our manuscript. In our response to main point 2 above, we explain that higher statistical power in low recombination regions is unlikely to explain our results alone, because we also show that the sweep deficit is substantially present not only in low recombination regions, but also requires the presence of a higher number of disease variants. We also describe in our response to main point 2 how our new MK-test results on long-term adaptation make it very unlikely that mendelian disease genes experience constitutively less adaptation. We want to thank the reviewer again for pointing out this issue with our manuscript, since it was indeed an important missing piece.

I'm a bit confused as to why the authors choose to promote the recessive deleterious hypothesis specifically. I think the result would be better supported if there were some additional source of information in to indicate that (a) this excess of recessive deleterious variants near disease associated genes exists, and (b) that the magnitude of the effect is consistent with the signals being reported.

We apologize for the confusion, and we hope that the added explanations as described above are sufficient. About point (a), Reviewer 1 provided the reference of Balick et al., 2015 Plos Genetics study that provides the evidence requested by the reviewer. We were not aware of this very valuable paper, but clearly should have been. About point (b), as detailed in our response to reviewer 1, we have now added population simulations that show that the observed magnitude of interference is consistent with the reported sweep deficits, as well as with the sweep deficit being much stronger in Africa than outside of Africa.

One part of this paper that was a bit difficult for me to understand at first was the sliding threshold scheme to summarize the strength of the evidence. The procedure is easy enough to understand, but it's not completely obvious at first how to interpret it statistically. After spending some time with it, here is how I understand it.The sliding threshold scheme amounts to an implicit upweighting of evidence from the genes with the strongest iHS signals, relative to those with lower signals, where the precise degree of upweighting depends on decisions about how many different thresholds to use. To see this, consider the simple case where we look at only the top 20 iHS signals, and we use just two rank thresholds, one at 10, and one at 20. Then, let us write D_{1:10} for the number of disease genes among the top 10 iHS(nSL) signals, and ND_{1:10} for the number of control non-disease genes among the top the top 10 iHS signals, and D_{1:20} and ND_{1:20} for the number of disease and non-disease genes among the top 20. The authors' summary statistic can be written asD_{1:10} – ND_{1:10} + D_{1:20} – ND_{1:20}Now, recognizing that D_{1:20} = D_{1:10} + D_{11:20} and ND_{1:20} = ND_{1:10} + ND_{11:20}, where D_{11:20} and ND_{11:20} give the number of disease and non-disease genes respectively among those with iHS(nSL) ranks 11-20, it follows that we can rewrite the authors' summary statistic asD_{1:10} – ND_{1:10} + (D_{1:10} + D_{11:20}) – (ND_{1:10} + ND_{11:20}) = 2*(D_{1:10} – ND_{1:10}) + 1*(D_{11:20} – ND_{11:20})which allows us to see the statistic as a sum of the difference in disease vs non-disease genes contained within non-overlapping bins, where each bin is weighted by a term that depends on how far down the list it is. In general, if there are K bins and we assign the first bin corresponding to genes that are below the first rank threshold and index of 1, the second bin, corresponding to genes below the second but above the first, an index of 2, etc., then the summary statistic is given bysum_k^K (K-K^+^1)*(D_k – ND_k)where I have slightly abused my notation by writing D_k for the number of disease genes in the ith bin (i.e. D_2 in my general expression corresponds to D_{11:20} in the specific example above).This formulation makes clearer (at least to me) how the analysis works by upweighting the contributions from more highly ranked iHS(nSL) bins, while still giving some weight to lower bins.In the text, the authors say that they do this to deliberately avoid an outlier approach, and that this approach "makes less assumptions on the expected strength of selective sweeps" (line 275/276). It's not obvious to me that this is true. Consider: for a particular distribution of "null" iHS(nSL) values at non-sweeps, and a particular distribution for "non-null" iHS(nSL) values at true sweeps, there is presumably some choice of weights that would be optimal for detecting the signal the authors are looking for. A standard outlier approach (e.g. look only at the top X iHS(nSL) signals) would amount to assigning equal weights to all of the bins ranked higher than X, and weights of zero to all other bins, while the authors' scheme amounts to a different set of weights that decay as we move from higher bins to lower bins, in a way that depends on exactly how many bins/thresholds are given, and how they are spaced. It is not obvious to me that the authors' scheme makes fewer assumptions, although it clearly makes different assumptions.I have written this out mostly to demistify the analysis pipeline to myself. Having done so, I do feel like I understand it better. I think that from a reader's perspective, the paper would be substantially improved by writing out the methods in clear symbolic notation. The authors don't necessarily have to include my new rewriting of their scheme in terms of non-overlapping bins (though I do think doing so would help clarify the relationship between their analysis and a more standard outlier approach), I think writing it down mathematically or perhaps writing out the algorithm in pseudo-code would help a lot.

We thank the reviewer for pointing out that we needed to be clearer about the statistic we use to estimate the false positive risk. Following their recommendation, we have now provided the requested definitions in symbolic notation. We wish to apologize for the confusion.

“We can write this FPR score as follows. With t being the number t threshold belonging to T, the set of rank threshold numbers, Dt the number of disease genes in sweeps at threshold number t, and C_t_ the number of control genes in sweeps at threshold number t then: […] This weighting scheme is justified, as it makes sense to give more weight to stronger, and therefore higher confidence sweep signals.”.

About the reviewer’s interrogation about whether or not our approach makes fewer assumptions than the classic outlier approach, it is true that the thresholds are still chosen arbitrarily. As mentioned by the reviewer at the start of their review, it is difficult to escape it. However, our approach does make less assumptions than the classic outlier approach, because we can get a significant result not only due to an enrichment of only the top, absolutely strongest sweeps, but also due for example to a large excess of weak or moderate sweeps, that would for example increase the expected numbers in the top 5,000 or top 2,000, without increasing the number of sweeps in the top 100 or top 50. Therefore, our approach is sensitive to a more diverse range of sweeps than the classic outlier approach, that makes a very restrictive assumption that sweeps have to be necessarily be strong.

We now explain this point better: “Using a range of rank thresholds makes less assumptions and provides more flexibility than the classic outlier approach, even though we still have to arbitrarily determine a list of rank thresholds to include. This is because we can get a significant result not only due to an enrichment of only the top, absolutely strongest sweeps, but also due for example to a large excess of weak or moderate sweeps, that would for example increase the expected numbers in the top 5,000 or top 2,000, without increasing the number of sweeps in the top 100 or top 50. Therefore, our approach is sensitive to a more diverse range of sweeps than the classic outlier approach, that makes a very restrictive assumption that sweeps have to be necessarily be strong.”.

It would be helpful if the authors could clarify earlier in the manuscript that their analysis is specifically going to focus on genes associated with Mendelian diseases. Currently, I believe this is not clarified until line 188, and as a reader, it left me feeling pretty unmoored throughout most of the introduction, because I couldn't figure out what "disease genes" really meant.

Here again, we want to apologize for the very avoidable confusion that ended up wasting the reviewers’ time. This was an avoidable mistake on our part, that we have now corrected with a revised title, introduction, and throughout the rest of the manuscript.

On line 250/251, the authors write:"All these factors have been shown to affect adaptation (Methods),"where "these factors" refers to a list given in the previous sentence. Here, the authors should provide a citation to previous work for this claim.

We apologize for the lack of precision of this sentence. We meant factors that could in principle affect adaptation, regardless of whether or not there are previous citations. We have now corrected the sentence accordingly. We thank the reviewer again for multiple important insights. Note that the factors that have been shown to affect adaptation have the corresponding citations in the Methods, when the confounding factors are introduced.

“All these factors have been shown to, or could in principle affect adaptation (Methods)”.